# Discovering the Active Ingredients of Medicine and Food Homologous Substances for Inhibiting the Cyclooxygenase-2 Metabolic Pathway by Machine Learning Algorithms

**DOI:** 10.3390/molecules28196782

**Published:** 2023-09-23

**Authors:** Yujia Tian, Zhixing Zhang, Aixia Yan

**Affiliations:** 1State Key Laboratory of Chemical Resource Engineering, Department of Pharmaceutical Engineering, Beijing University of Chemical Technology, 15 Beisanhuan East Road, Beijing 100029, China; 2020400253@mail.buct.edu.cn (Y.T.); y20220085@mail.ecust.edu.cn (Z.Z.); 2Shanghai Key Laboratory of New Drug Design, School of Pharmacy, East China University of Science and Technology, 130 Meilong Road, Shanghai 200237, China

**Keywords:** medicine and food homology (MFH), machine learning, ensemble learning, virtual screening, anti-inflammation

## Abstract

Cyclooxygenase-2 (COX-2) and microsomal prostaglandin E_2_ synthase (mPGES-1) are two key targets in anti-inflammatory therapy. Medicine and food homology (MFH) substances have both edible and medicinal properties, providing a valuable resource for the development of novel, safe, and efficient COX-2 and mPGES-1 inhibitors. In this study, we collected active ingredients from 503 MFH substances and constructed the first comprehensive MFH database containing 27,319 molecules. Subsequently, we performed Murcko scaffold analysis and K-means clustering to deeply analyze the composition of the constructed database and evaluate its structural diversity. Furthermore, we employed four supervised machine learning algorithms, including support vector machine (SVM), random forest (RF), deep neural networks (DNNs), and eXtreme Gradient Boosting (XGBoost), as well as ensemble learning, to establish 640 classification models and 160 regression models for COX-2 and mPGES-1 inhibitors. Among them, ModelA_ensemble_RF_1 emerged as the optimal classification model for COX-2 inhibitors, achieving predicted Matthews correlation coefficient (MCC) values of 0.802 and 0.603 on the test set and external validation set, respectively. ModelC_RDKIT_SVM_2 was identified as the best regression model based on COX-2 inhibitors, with root mean squared error (RMSE) values of 0.419 and 0.513 on the test set and external validation set, respectively. ModelD_ECFP_SVM_4 stood out as the top classification model for mPGES-1 inhibitors, attaining MCC values of 0.832 and 0.584 on the test set and external validation set, respectively. The optimal regression model for mPGES-1 inhibitors, ModelF_3D_SVM_1, exhibited predictive RMSE values of 0.253 and 0.35 on the test set and external validation set, respectively. Finally, we proposed a ligand-based cascade virtual screening strategy, which integrated the well-performing supervised machine learning models with unsupervised learning: the self-organized map (SOM) and molecular scaffold analysis. Using this virtual screening workflow, we discovered 10 potential COX-2 inhibitors and 15 potential mPGES-1 inhibitors from the MFH database. We further verified candidates by molecular docking, investigated the interaction of the candidate molecules upon binding to COX-2 or mPGES-1. The constructed comprehensive MFH database has laid a solid foundation for the further research and utilization of the MFH substances. The series of well-performing machine learning models can be employed to predict the COX-2 and mPGES-1 inhibitory capabilities of unknown compounds, thereby aiding in the discovery of anti-inflammatory medications. The COX-2 and mPGES-1 potential inhibitor molecules identified through the cascade virtual screening approach provide insights and references for the design of highly effective and safe novel anti-inflammatory drugs.

## 1. Introduction

Eicosanoids, derived from arachidonic acid (AA) and related polyunsaturated fatty acids (PUFAs), play a crucial role in regulating a wide range of homeostatic and inflammatory processes associated with many diseases, such as atherosclerosis, Alzheimer’s disease, and cancer [1]. Studies of eicosanoids have primarily focused on prostaglandin E2 (PGE_2_), due to the fact that PEG_2_ primarily contributes to the fundamental symptoms of inflammation, such as fever, swelling, redness, pain, and loss of function [2]. In the production of PGE_2_, cyclooxygenase-2 (COX-2) catalyzes the conversion of arachidonic acid (AA) into prostaglandin H2 (PGH_2_). Subsequently, microsomal prostaglandin E2 synthase (mPGES-1) converts PGH_2_ into PGE_2_. Therefore, suppressing the overexpression of COX-2 and mPGES-1 is a reasonable strategy in the treatment of inflammation.

Traditional non-steroidal anti-inflammatory drugs (NSAIDs), such as Aspirin, inhibit both COX-1 and COX-2. This reduces the production of the pro-inflammatory factor PEG_2_ along with the cytoprotective prostaglandin I_2_, leading to an increased risk of severe gastrointestinal and cardiovascular disease. Although highly selective COX2 inhibitors, such as Celecoxib and Rofecoxib, can alleviate some of the side effects associated with NSAIDs, their long-term use does not completely avoid the risk of cardiovascular disease [3]. Consequently, current research is focused on discovering potential COX-2 inhibitors that exhibit both high selectivity and low toxicity. In recent years, mPGES-1 inhibitors have been suggested as attractive anti-inflammation therapeutics since they can selectively reduce the PEG_2_ without affecting the cytoprotective PGs that regulate homeostasis. However, the wide variation in mPGES-1 inhibitor efficacy between humans and mice poses a challenge to fully exploit rodent disease models, and the lack of effective inhibitors with cross-species activity further hinders the development of safe and effective mPGES-1 inhibitors [4].

Medicine and food homology (MFH) refers to substances that are both food and herbal medicine. It reflects the traditional Chinese idea of health care, and includes the contents of food therapy, health care, and medicinal food in Chinese medicine [5]. MFHs have passed the food safety risk assessment of the Chinese National Health Commission and have been proven to be both for long-term use as daily food and for disease treatment and health benefits [6]. Natural products have served as a valuable source of drugs and lead compounds due to their complexity, structural diversity, and extensive pharmacological effects. Growing studies have revealed that many active ingredients of plant-derived and marine-derived natural products are involved in the regulation of inflammatory responses in humans [7], especially with COX-2 and mPGES-1 inhibitory activity. A series of secondary metabolites isolated from marine products have shown potency in inhibiting COX-2, their chemical structures are shown in sub-figure A of Figure 1. Vaccinal A, Botryoisocoumarin A [8], Axinelline A, Capnellene, Stachybogrisephenone B, and Actinoquinoline A inhibit COX-2 in vitro with IC_50_ values of 1.8 μM, 6.5 μM, 2.8 μM, 6.2 μM, 8.9 μM, and 2.1 μM, respectively [9]. Some compounds from medicinal plants have been verified to exhibit the prominent mPGES-1 inhibitory effect (seen in sub-figure B of Figure 1), Curcumin, Hyperforin, Garcinol, Arzanol, Myrtucommulone, Carnosic acid, and Emeblin inhibit mPGES-1 in vitro with IC_50_ values of 0.3 μM, 1 μM, 0.3 μM, 0.4 μM, 1 μM, 5 μM, and 0.2 μM, respectively [10]. Although many natural products are reported to be potent COX-2, mPGES-1 inhibitors or transcriptional suppressors, poor oral availability and non-negligible toxicity still limit their further clinical use. MFHs are easily absorbed and utilized by humans with relative nontoxicity. Additionally, there are some multitarget anti-inflammatory compounds in Chinese herbal medicine. Acrovestone, extracted from acronychia, inhibits mPGES-1 and 5-lipoxygenase (5-LOX) in vitro with IC_50_ values of 2.7 μM and 1.1 μM, respectively [11]. Cannflavin A, extracted from Cannabis sativa, inhibits mPGES-1 and 5-LOX in vitro with IC_50_ values of 1.8 μM and 0.9 μM, respectively. The above multi-target inhibitors can reduce the risk of drug–drug interactions and contribute to efficient anti-inflammatory therapy [12]. Therefore, MFHs are capable of providing insight into the discovery of novel, safe, and effective COX-2 and mPGES-1 inhibitors.

There has been a growing interest in the application of machine learning (ML) and deep learning (DL) in the field of biomedicine. Various emerging supervised and unsupervised learning algorithms have shown potential in providing valuable insights and predictions for drug discovery, drug repurposing, diagnostics, and pharmaceutical production [13]. Currently, virtual screening (VS) combined with experimental verification is the standard protocol for drug discovery. Some ML-facilitating VS approaches have been widely used to mine novel molecular substances from in-house or commercial compound libraries [14]. An ML-based Quantitative structure–activity relationship (QSAR) model has become a favorable tool in VS due to its ability to rapidly output predictions based on input datasets and its high hit rate [15]. Recent studies have focused on implementing VS on natural compound libraries; this is because active ingredients derived from natural products are considered an ideal starting point for designing anti-inflammatory agents [16]. Wang’s group [17] built several classification models and employed these models for identifying potential P-glycoprotein inhibitors in the Traditional Chinese Medicine Systems Pharmacology (TCMSP) database. Sattar et al. [18] utilized molecular docking to screen potential COX inhibitors on compounds isolated from *Eucalyptus maculata* resin, and they found that 1,6-dicinnamoyl-O-α-D-glucopyranoside exhibited an COX-2 inhibitory effect.

Currently, obscure functions and unclear therapeutic targets limit the clinical practice of medicine and food homology (MFH) substances. In this study, we constructed the first comprehensive MFH database aiming at further investigating the complex active ingredients of MFH and deciphering its biology and functional activities (detailed workflow is shown in Figure 2). Additionally, we constructed the updated comprehensive dataset of COX-2 and mPGES-1 inhibitors. Leveraging these datasets, we employed supervised machine learning algorithms and ensemble learning techniques to develop a series of QSAR models for predicting the inhibitory efficacy of COX-2 and mPGES-1. Furthermore, utilizing the well-performed machine learning models combined with unsupervised learning and scaffold analysis, we performed virtual screening on the MFH database to identify potential COX-2 and mPGES-1 inhibitors which could contribute to the efficient and safe anti-inflammatory therapy.

## 2. Results and Discussion

### 2.1. Chemical Space and Scaffold Analysis of the MFH Database

We constructed a database containing 27,319 active ingredient molecules derived from 503 kinds of MFH substances by collecting data from the Traditional Chinese Medicines Integrated Database (TCMID) [19] and Traditional Chinese Medicine Systems Pharmacology Database and Analysis Platform (TCMSP) [20], as well as 92 pieces of literature. Details of the constructed MFH database can be seen in Appendix A. The distribution of two physicochemical properties of the MFH database is shown in Figure 3. The molecular weight (MW) of molecules derived from TCMID, TCMSP, and the literature ranges from 32 to 1900, 30 to 1466, and 59 to 1263, respectively. The octanol–water partition coefficient (LogP) of molecules derived from TCMID, TCMSP, and the literature ranges from −24.9 to 24, −11.9 to 24.5, and −8.8 to 12.6, respectively. The distribution of basic physicochemical properties across two orders of magnitude demonstrates the extensive chemical space of our MFH database.

To further evaluate the structural diversity of our MFH database, we calculated the Tanimoto coefficients (TCs) [21] on molecules (represented by 1024 bits ECFP4 fingerprints). The average TC value for the entire MFH database was 0.216, with 95.92% of molecule pairs having TC values of less than 0.6, indicating significant structural differences within our MFH database.

In addition, we extracted Murcko scaffolds of the MFH database and performed a K-means clustering [22] analysis. As a result, the MFH database was divided into 11 clusters. The Flavonoids cluster was the most abundant, comprising approximately one-fifth of the database, while Fatty Acids, Saponins, and Steroids each accounted for roughly one-tenth. Moreover, there were Lignans, Alkaloids, Triterpenoids, Sesquiterpenes, Diterpenes, and Stilbenes in the MFH database (seen in Figure 4A). These are commonly isolated components from natural products, and some Flavonoids, Steroids, Alkaloids, and Triterpenoids have been reported to play important roles in the regulation of immune responses [23]. We further summarized the top 20 Murcko scaffolds in the MFH database, as shown in Figure 4B; the scaffolds ranged from simple aromatic natural products with a single ring to complex skeletons with 7–8 ring systems. These observations demonstrate that our MFH database is comprehensive and exhibits high structural diversity.

### 2.2. Performances Evaluation and Comparison of Developed Models

In this study, we developed classification models using four datasets: Dataset 1 and 2, which contain COX-2 inhibitors, and Dataset 4 and 5, which are composed of mPGES-1 inhibitors. To generate training and test sets, we split the datasets for classification 10 times randomly. We then characterized these sets using three types of fingerprints: Avalon, ECFP4, and MACCS. Four machine learning methods were applied to predict high or weak inhibition on COX-2 and mPGES-1 separately. As a result, we constructed a total of 480 classification models. Subsequently, we integrated the predicted probabilities of classification models (built with the same algorithm but with the different molecular characterization) by a stacked generalization approach. This resulted in the construction of an additional 160 ensemble classification models: 80 for COX-2 inhibitors and 80 for mPGES-1 inhibitors. Additionally, we utilized two datasets, Dataset 3 and Dataset 6, represented by two types of descriptors to build 160 regression models following a similar process of developing classification models.

#### 2.2.1. Performances of Classification and Ensemble Models on Dataset 1

The Matthews correlation coefficients (MCCs) on the test sets were employed to evaluate the predictive ability and stability of constructed classification models, and the MCC on external validation sets was applied to assess the generalization ability of models. Table 1 lists the overall model performances based on Dataset 1 (1640 COX-2 inhibitors), and the detailed performances of all models on Dataset 1 can be seen in Appendix A. Figure 5A depicts the MCC values of 10 randomly split test sets for Dataset 1.

When comparing the performance of different algorithms, SVM and XGB slightly outperformed other machine learning methods. When using Avalon fingerprints to characterize the dataset, the mean MCC values on 10 randomly split test sets for SVM and XGB models were 0.783 and 0.744, respectively. When using ECFP fingerprints, the mean MCC values for the SVM and XGBoost models were 0.763 and 0.768, respectively. When using MACCS fingerprints, the mean MCC values for the SVM and XGBoost models were 0.746 and 0.753, respectively. Meanwhile, the RF algorithm had a slightly inferior performance with the average MCC values on 10 randomly split test sets of 0.721, 0.704, and 0.701 when using Avalon, ECFP, and MACCS fingerprints to characterize the dataset, respectively.

When comparing the performance of different descriptors, all three fingerprints (Avalon, ECFP, and MACCS) were equally effective in characterizing the dataset with no significant differences. The MCC values on test sets for all models exceeded 0.66. Compared to the base classification models, the ensemble models that integrated all three fingerprints showed improved performance with mean MCC values of 0.735, 0.782, 0.747, and 0.774 on test sets when using RF, SVM, DNN, and XGBoost models, respectively. The ensemble model for each algorithm had an average MCC improvement of 0.02 on the test set compared to the models constructed with a single type of fingerprints.

#### 2.2.2. Performances of Classification and Ensemble Models on Dataset 2

The overall performances of developed classifiers on Dataset 2 were summarized in Table 2, the detailed performances of all models on Dataset 2 (containing 2925 COX-2 inhibitors) can be seen in Appendix A. The detailed MCC values of 10 randomly split test sets for Dataset 2 were visualized in Figure 5B. The MCC values of the test sets of the models on dataset 2 all exceeded 0.52, which was a decrease in performance compared to the classification models constructed with Dataset 1. This was mainly because the moderately active inhibitors were removed from dataset 1, and the structural differences between the highly/weakly inhibitors were easily distinguished. Similar to the performances on Dataset 1, the SVM and XGBoost algorithms also exhibited excellent performance for Dataset 2, with MCC values of 0.632 ± 0.018, 0.63 ± 0.022, and 0.621 ± 0.025 for the SVM model on the test set of the dataset characterized using Avalon, ECFP, and MACCS, respectively. The MCC values of the XGBoost model on the test set of the dataset represented by Avalon, ECFP, and MACCS were 0.644 ± 0.014, 0.625 ± 0.028, and 0.626 ± 0.012, respectively. The MCC performances of the ensemble RF, SVM, DNN, and XGBoost models on the test sets were 0.576 ± 0.02, 0.642 ± 0.016, 0.604 ± 0.015, and 0.643 ± 0.012, respectively. Compared with models constructed with one type of fingerprints, the ensemble models of each algorithm increased the average MCC on the test set by 0.01.

#### 2.2.3. Performances of Regression Models on Dataset 3

Table 3 displays the overall performances of the models using two criteria (R^2^ and RMSE) based on Dataset 3, which consisted of 1511 COX-2 inhibitors. Detailed results of all 80 QSAR models are listed in Appendix A. The RMSE based on the test sets and external validation sets were utilized as the main metric to evaluate the performance of the regression models. Figure 6 showed the specific RMSE values of 10 randomly split test sets on Dataset 3.

The RMSE values for the test set of models constructed using the G_3D descriptors were all below 0.56. From the perspective of modeling algorithms, the SVM algorithm performed the best with an average RMSE of 0.465 on the test sets. Among the models constructed using RDKit descriptors, the SVM algorithm also showed the highest predictive power with a mean RMSE of 0.436 ± 0.019, followed by the comparable DNN and XGBoost algorithms with mean RMSE values of 0.471 ± 0.019, respectively. Comparing from the descriptor perspective, the RMSEs of the test sets of the models constructed using RDKit descriptors were all below 0.54. Based on the same algorithm, RDKit-based models performed slightly better than the G_3D-based models, which indicates that the RDKit 2D descriptor is more suitable for characterizing the COX-2 inhibitors collected in this study.

#### 2.2.4. Performances of Classification and Ensemble Models on Dataset 4

Table 4 summarizes the overall model performances of Dataset 4, which included 3179 mPGES-1 inhibitors. The detailed performances of all models on Dataset 4 can be seen in Appendix A and Figure 7A. The XGBoost algorithm performed the best, with mean MCC values of 0.74, 0.83, and 0.765 for 10 randomly partitioned test sets based on the Avalon, ECFP4, and MACCS characterized datasets, respectively. Comparing from the perspective of dataset characterization, ECFP4 fingerprints slightly outperformed the other two fingerprints. The highest MCC values were obtained on the test sets of the SVM, DNN, and XGBoost models using the dataset characterized by ECFP4 fingerprints. Among the ensemble classification models constructed based on the three fingerprints, the ensemble SVM and XGBoost models achieved excellent MCC performances of 0.783 ± 0.017 and 0.788 ± 0.006 on the test sets, respectively.

#### 2.2.5. Performances of Classification and Ensemble Models on Dataset 5

Compared to the classification models from Dataset 4 (Table 4), the classification models constructed based on Dataset 5, containing 3455 mPGES-1 inhibitors, were slightly less discriminative for highly/weakly active mPGES-1 inhibitors, shown in Table 5, Appendix A, and Figure 7B. In terms of machine learning algorithms, the models established by the RF algorithm had the worst performance, with all the average MCC values below 0.6 on the test set based on the datasets characterized by three types of fingerprints. Conversely, the SVM and XGBoost algorithms exhibited better performance. XGBoost models achieved the best performance on the test set based on the datasets characterized by Avalon and MACCS. On the test set based on the ECFP4 characterization, the SVM and XGBoost models achieved excellent performance with average MCC values of 0.773 and 0.772, respectively. In addition, the ensemble SVM and XGBoost classification models obtained average MCC values of 0.738 ± 0.017 and 0.749 ± 0.011 on the test sets, respectively.

#### 2.2.6. Performances of Regression Models on Dataset 6

Based on Dataset 6 consisting of 735 mPGES-1 inhibitors, a total of 80 regression models (Table 6) were constructed using the four algorithms to predict the bioactivities of the inhibitors; the details of all model performances are listed in Appendix A. As seen in Table 6 and Figure 8, the RMSE values of the test sets of the models constructed using the G_3D descriptors were all below 0.42, with the SVM algorithm performing the best, with an average RMSE of 0.329 on the test sets. This was followed by models constructed by the XGBoost algorithm, with an average RMSE of 0.339 on the test sets. The RMSE values of the test sets of the models constructed using the 2D RDKit descriptors were all below 0.35. From the descriptors’ perspective, RDKit-based models performed slightly better than G_3D-based models for the same algorithm, with an average RMSE reduction of 0.03 in the test sets.

#### 2.2.7. Performances of the External Validation Sets

In addition to evaluating the predictive accuracy and stability of the models through the performance of the five-fold cross-validation and test sets, four external validation sets were also employed to verify the generalization ability of the constructed classification and regression models. Only models that performed well on the external validation sets were applied to further implement virtual screening on the MFH database. The ensemble SVM models and the ensemble XGBoost models performed best in the classification models constructed based on Dataset 1, with predicted MCC values of 0.551 ± 0.012 and 0.546 ± 0.014 for external validation set A1, respectively. Similarly, the ensemble SVM and the ensemble XGBoost models outperformed others in the classification models developed based on Dataset 2, with predicted MCC values of 0.544 ± 0.011 and 0.531 ± 0.01, respectively. For the regression model built on Dataset 3, the regression models constructed using the XGBoost algorithm and the RDkit descriptors had the most accurate prediction RMSE values of 0.584 ± 0.035 for external validation set A2. In the classification model constructed based on Dataset 4, models constructed by the XGBoost with the ECFP4 fingerprints had the highest prediction for external validation set B1 with an MCC of 0.547 ± 0.015. For classification models established based on Dataset 5, the model developed using SVM and XGBoost algorithms with ECFP4 fingerprints had the best performance; the predicted MCC values for external validation set B1 reached 0.538 ± 0.016 and 0.537 ± 0.009, respectively. Models constructed through the SVM algorithm combined with RDkit descriptors outperformed the regression models built based on Dataset 6, with a predicted RMSE of 0.424 ± 0.019 for external validation set B2. The specific model results and parameters applied for virtual screening of MFH database are summarized in Appendix A.

### 2.3. Virtual Screening on the MFH Database

In this study, we employed a ligand-based cascade virtual screening strategy to predict potential COX-2 and mPGES-1 inhibitors from the constructed MFH database containing 27,319 molecules. Our goal was to advance the development of safer anti-inflammatory therapies. The virtual screening workflow proceeded as follows, taking the screening of COX-2 inhibitors as an example. Firstly, molecules of the MFH database were predicted using 39 classification models that exhibited excellent performance on external validation sets (seen in Appendix A). These 39 classification models comprised both single classifiers and ensemble classifiers constructed based on Dataset 1, as well as basic classifiers and ensemble classifiers constructed based on Dataset 2. By combining different datasets with multiple fingerprints and employing various supervised machine learning algorithms and the ensemble algorithm, the classification models used to predict the COX-2 inhibitory activity of molecules in the MFH database demonstrated strong generalization ability and robustness, thus providing reliability to the prediction results to some extent. Subsequently, the molecules predicted as highly active inhibitors by the 39 classification models were further input into 17 regression models to predict their specific inhibitory values against COX-2 (Detailed performances are listed in Appendix A). The predicted values from the 17 regression models were averaged to mitigate occasional errors introduced by individual models, further enhancing the credibility of the prediction results. Based on the ranked mean predicted bioactivities from the 17 regression models, molecules with an average predicted IC_50_ values below 10 μM (a common threshold for inhibitory potency) were retained. Additionally, we applied the self-organized map (SOM), an unsupervised algorithm, to predict the molecules of the MFH database. The positioning of molecules on a two-dimensional neural network was used to determine their high or weak COX-2 inhibitory activity. This approach enabled the retention of MFH molecules that resided in the same location as the majority of highly active molecules (over 80%). Finally, COX-2 inhibitor candidates were selected through an extensive literature review and molecular scaffold analysis. The screening process for mPGES-1 inhibitors followed a similar procedure, as described above.

#### 2.3.1. Potential COX-2 Inhibitors in the MFH Database

Through the cascade virtual screening described above, 10 potential COX-2 inhibitors were selected from a pool of 27,319 MFH molecules. The structures of these molecules, along with their positions on the two-dimensional grid mapped by SOM, were presented in Figure 9. From Figure 8, it can be observed that the majority of the MFH molecules are distributed in different grid cells compared to known COX-2 inhibitor molecules. This discrepancy arises due to significant structural differences between natural MFH molecules and the synthesized or modified COX-2 inhibitors. Our focus primarily lies on the MFH molecules that share the same position as reported natural COX-2 inhibitors and those located in the grids predominantly occupied by highly active inhibitors (rendered in red in Figure 9). HP represents the proportion of the grid occupied by highly active inhibitors. It is the proportion of the number of highly active inhibitors in the grid to all molecules in the grid. The larger the calculated HP of one grid (close to 1), the warmer the color tends to be (red). On the contrary, the closer the HP value of a grid is to 0, the cooler the color tends to be (blue). The origin, predicted bioactivity values, and reported effects of the candidate molecules are summarized in Table 7.

Candidate cmp_A3, also known as Humulene, is derived from *Panax Ginseng* and belongs to the monocyclic sesquiterpene. It has been reported to exhibit inhibitory activity against COX-2, suppressing the expression of COX-2 in mice and reducing the production of prostaglandin E_2_ (PGE_2_) [24]. Another candidate, cmp_A1 (Dehydrotanshinone II A), derived from *Radix Salviae*, is the benzofuran derivatives. Kwon et al. isolated Dehydrotanshinone II A from *Salvia miltiorrhiza* Bunge and validated its inhibitory effect on COX-2 using a platelet activation model [25]. Candidate cmp_A6 (β-Sesquiphellandrene) is commonly found in *Atractylodes macrocephala* and *Zingiber officinale*. *Zingiber officinale* oil has been reported to possess anti-inflammatory activity, particularly by inhibiting lipoxygenase [26]. cmp_A7, derived from *Lycii Fructus*, is an apocarotenoid that significantly inhibits the expression of IL-1β in vitro, thus exerting an anti-inflammatory effect [27]. Cmp_A5, named Scutianine C, is derived from in *Jujubae Fructus* and has been reported as a biologically active alkaloid with antimicrobial properties [28]. Candidate cmp_A10 (Hispaglabridin B) isolated from *Glycyrrhiza glabra* L. belongs to the isoflavone derivatives and is located in the same grid as the reported natural COX-2 inhibitors, which has demonstrated antioxidant effects [29]. Peptidomimetic candidates, cmp_A2 and cmp_A8, originate from *Corneum Gigeriae Galli Endothelium* and *Fagopyrum esculentum*, respectively. They were predicted by SOM to have a probability of 0.821 as high-activity COX-2 inhibitors, with average predicted IC_50_ values of 2.69 μM and 5.2 μM, respectively. Candidate cmp_A4, classified as a sesquiterpene, is derived from *Angelica sinensis Radix* and was predicted by SOM as a 100% high-activity COX-2 inhibitor, with an average predicted IC_50_ value of 3.69 μM. cmp_A9, found in *Mori Follum*, is a folinic acid with an average predicted IC_50_ value of 5.92 μM.

**Table 7 molecules-28-06782-t007:** The COX-2 inhibitors candidates from the MFH database found by virtual screening using the related ML models built in this work.

Candidates	Origins	IC_50_ μM ^a^	SOM HP ^b^	Effects
cmp_A1	*Radix Salviae*	1.74	0.91	Anti-inflammation [25]
cmp_A2	*Gigeriae Galli Endothelium*	2.69	0.82	
cmp_A3	*Panax Ginseng*	3.47	1	COX-2 inhibition [24]
cmp_A4	*Angelica sinensis Radix*	3.69	1	
cmp_A5	*Jujubae Fructus*	4.24	1	Antibacterial [28]
cmp_A6	*Atractylodes macrocephala*	4.37	0.91	Anti-inflammation [26]
cmp_A7	*Lycii Fructus*	4.60	0.91	Anti-inflammation [27]
cmp_A8	*Fagopyrum esculentum*	5.20	0.82	
cmp_A9	*Mori Follum*	5.92	0.82	
cmp_A10	*Glycyrrhiza glabra* L.	6.97	1	Anti-oxidation [29]

^a^: the average predicted IC_50_ values by a series of optimal regression models. ^b^: the percentage of highly active inhibitors mapped to the target molecules in the same position during the self-organized map (SOM) training. It can also be taken as the probability of being predicted by an unsupervised algorithm as a highly active COX-2 inhibitor.

#### 2.3.2. Potential mPGES-1 Inhibitors in the MFH Database

Through the virtual screening process described above, 15 potential mPGES-1 inhibitors were screened from the MFH database. As seen in Figure 10 and Table 8, Candidate cmp_B1, a cannabinolic acid derived from *Cannabis Sativa* L., was mapped by SOM into a grid inhabited by all highly active mPGES-1 inhibitors. The average predicted IC_50_ value from the regression models for cmp_B1 was 0.88 μM. Candidate cmp_B1 has been reported to decrease COX enzyme activity, although selectivity toward COX1/2 was still unknown [30]. Candidate cmp_B2, isolated from *Ramulus Mori*, is a prenylated flavanone. It was predicted by the SOM to be a highly active mPGES-1 inhibitor with a probability of 0.93 and a mean IC_50_ of 0.25 μM by the regression models. Candidate cmp_B3, an active ingredient of *Amomum longiligularg*, is an isopentenyl flavonoid with a predicted IC_50_ of 0.34 μM, and it has been reported to inhibit the growth of breast cancer cells in vitro and in vivo [31]. Candidate cmp_B4 is also known as Kanzonol C, which is a flavonoid-like active ingredient of *Glycyrrhiza glabra* L. It has been demonstrated as a PTP1B inhibitor [32] and has been found to exhibit inhibitory activity against nitric oxide (NO), making it a potential anti-inflammatory agent [33]. Candidate cmp_B5, derived from *Gardeniae Fructus*, is a caffeoylquinic acid with a predicted IC_50_ value of 0.18 μM. It possesses the ability to inhibit lipoxygenase [34]. Candidate cmp_B6 is the active ingredient of *Schisandra chinensis* with an average predicted IC_50_ value of 0.37 μM, which has been reported to inhibit UDP-glucuronosyltransferase [35] and Oxidized low-density lipoprotein (OxLDL) [36]. Candidate cmp_B7, also known as Garcinone B, derived from *Rhizoma Dioscoreae* (Chinese yam), has a predicted IC_50_ value of 0.55 μM. It has been found to reduce the production of prostaglandin E_2_, although the specific metabolic pathway of its action has yet to be demonstrated [37]. Candidate cmp_B8 (Kuwanon M) was isolated from *Ramulus Mori*. It was predicted by the SOM to be a highly active mPGES-1 inhibitor with a probability of 1 and was predicted by the regression models to have a mean IC_50_ of 0.18 μM. Candidate cmp_B9, derived from *Mori Cortex*, is an isopropenylated phenol derivative that has been reported to show an inhibitory effect on Tyrosinase [38]. Candidate cmp_B10, derived from *Glehniae Radix*, is a coumarin analogue with a predicted IC_50_ value of 0.2 μM. Candidate cmp_B11, also known as Xanthochymol, is a component of *Colla* and belongs to the polycyclic phloroglucinol. It has been found to regulate inflammation by downregulating the expression of several major histocompatibility complex (MHC) molecules [39]. Candidate cmp_B12, derived from *Epimrdii Herba*, is a flavonoid with a predicted IC_50_ value of 0.5 μM. It has been reported to inhibit CYP3A4, thus exhibiting anti-inflammatory effects [40]. Candidate cmp_B13, isolated from *Coicis Semen*, is a steroid with an average predicted IC_50_ value of 0.27 μM. Lee et al. [41] verified the inhibitory effect of γ-oryzanol, composed of cmp_B13 and various ferulate molecules, on inflammation-related diseases. Candidate cmp_B14, also known as Kaikasaponin III, derived from *Radix Puerariae*, is a triterpenoid with a predicted IC_50_ value of 0.2 μM. It possesses antioxidant effects [42] and has been shown to have therapeutic effects on colitis [43]. Candidate cmp_B15, located in the same grid as cmp_B14, is also a triterpenoid primarily found in *Alisma Orientale*. It has a predicted IC_50_ value of 0.29 μM and has been reported to inhibit COX-2 expression [44].

#### 2.3.3. Molecular Docking on the Potential COX-2 and mPGES-1 Inhibitors

Through a series of ligand-based virtual screening processes, we have identified potential inhibitors of COX-2 and mPGES-1 from the established MFH database, and we further filtered these candidate molecules by the pan assay interference compounds (PAINS) rule [45]. To further validate the reliability and validity of the virtual screening, we conducted molecular docking (a widely utilized structure-based virtual screening approach) on the candidate MFH molecules. This procedure aimed to examine the binding modes between the candidate molecules and the proteins, while also assessing the ability of the candidate molecules to interact with the key amino acids of the target protein.

##### Molecular Docking Analysis on Potential COX-2 Inhibitors

The active site of COX-2 is demarcated from the initial substrate binding site by a constriction formed by three residues: Arg120, Tyr355, and Glu524. This structural constriction necessitates dilation to enable the entry or exit of substrates to and from the active site [46]. Ser530 and Tyr385 are also vital during the catalytic process of COX-2. Several typical binding modes of non-steroidal anti-inflammatory drugs (NSAIDs) interacting with COX-2 have been reported. These modes include the hydrophobic region of NSAIDs interacting with Tyr385, Trp387, and neighboring residues; the polar region of NSAIDs interacting with residues located above Tyr355; NSAIDs with negative charges interacting with Arg120 [47].

We calculated the structural similarity (measured by Tanimoto coefficient) between the candidate MFH molecules and the ligand molecules in the published crystal structures, we selected the protein crystal structure with the bound ligand having the highest structural similarity to the candidate MFH molecules for docking, the details of which are shown in Appendix A. As shown in Table 9, candidate compounds cmp_A3, cmp_A4, cmp_A6, and cmp_A7 bind within the active cavity of COX-2 (PDB: 4PH9), with binding affinities of −7.37, −8.53, −8.68, and −7.69 kcal/mol, respectively. The co-crystallized ligand, ibuprofen, in the complex 4PH9, under the same docking conditions, exhibits a binding affinity of −9.41 kcal/mol. While the four candidate molecules displayed affinities lower than the original ligand within the protein, they all established interactions with key amino acid residues in the catalytic domain of COX-2: hydrophobic interactions with Trp388 and polar interactions involving Tyr356, Ser531, and Tyr386. Candidate cmp_A7 generated a hydrogen bond interaction with Arg121, indicating a robust binding force. Candidate cmp_A1 bound within the active pocket of COX-2 (PDB: 5KIR), exhibiting a binding affinity of −7.57 kcal/mol. Meanwhile, the co-crystallized ligand within complex 5KIR achieved a binding affinity of −9.80 kcal/mol under identical docking conditions. Candidate cmp_A3 engaged in polar interactions with Tyr355 and formed a π-H stacking interaction with Ser533. Candidate cmp_A2 and cmp_A10 occupied the active site of the protein with PDB index 6BL3, demonstrating binding affinities of −11.75 and −8.92 kcal/mol, respectively. The co-crystallized ligand within complex 6BL3 had a binding affinity of −12.15 kcal/mol under the same docking procedure. Candidate cmp_A2 generated hydrogen bond interactions with Ser530, Lys83, and Glu524, engaged in hydrophobic interactions with Trp387, and established polar interactions with Tyr355 and Tyr385. Candidate cmp_A10 formed hydrogen bond interactions with Glu524, had hydrophobic interactions with Trp387, and engaged in polar interactions with Ser530 and Tyr355. Candidates cmp_A5, cmp_A8, and cmp_A9 bound to the active cavity of COX-2 (PDB: 6BL4) with the affinities of −7.25, −11.24, and −9.61 kcal/mol, respectively. The affinity of the complex 6BL4 after docking with the original ligand was −12.27 kcal/mol. Candidate cmp_A5 engaged in hydrophobic interactions with Trp100 and formed polar interactions with Tyr115, Lys79, and Lys83. Candidate cmp_A8 established hydrogen bond interactions with Tyr355, Arg120, Glu524, Ser530, and Lys83, which significantly contributed to its enhanced protein affinity. Candidate cmp_A9 had hydrogen bond interactions with Ser119, Met522, Glu524, and Ser530.

In summary, the candidate MFH molecules could effectively bind to the active pocket of COX-2 and interact with key amino acids involved in COX-2 catalysis. These interactions were consistent with the classical binding interactions reported between NSAIDs and COX-2. Most candidate MFH molecules exhibited strong hydrogen bonding interactions with the key amino acid residues of COX-2. These observations support the reliability of the potential COX-2 inhibitor molecules discovered through the ligand-based virtual screening on the MFH database.

##### Molecular Docking Analysis on Potential mPGES-1 Inhibitors

mPGES-1 is a homotrimer, with each subunit consisting of four transmembrane helices. The mPGES-1 trimer contains three active site cavities, which are formed collectively by transmembrane helices 1, 2, and 4 along with neighboring monomers [48]. Among the currently resolved co-crystal complexes, key amino acids include AlaA123, ProA124, SerA127, ValA128, TyrA130, ThrA131, GlnA134, TyrB28, IleB32, AsnB36, ArgB38, LeuB39, PheB44, ArgB52, and HisB53, where A and B representing different monomers in the mPGES-1 trimer, respectively.

The selection of mPGES-1 crystals for docking was carried out using a similar methodology as the COX-2 crystal selection, with the detailed results presented in Appendix A. The binding affinities and interactions between candidate MFH molecules and mPGES-1 are summarized in Table 10. Candidate cmp_B4 and cmp_B5 bound within the active pocket of the protein 4AL1, exhibiting binding affinities of −8.08 and −11.25 kcal/mol, respectively. The binding affinity of the original ligand of protein 4AL1 is −12.12 kcal/mol. Specifically, Candidate cmp_B4 formed hydrogen bond interactions with AspB49, AsnB46, and ThrA131. Candidate cmp_B5 established hydrogen bond interactions with AsnB46, ArgB52, GluA77, and TyrA117, with ionic bonds between ArgB38 and ArgA126, and π-H stacking with Ile B32. Candidate cmp_B1, cmp_B6, cmp_B7, cmp_B11, cmp_B12, cmp_B13, cmp_B14, and cmp_B15 were docked within the active site of the protein 4YL0, exhibiting binding affinities of −7.93, −7.25, −7.26, −6.89, −8.73, −7.19, −10.71, and −7.34 kcal/mol, respectively. The binding affinity of protein 4YL0 after docking with its ligand is −7.17 kcal/mol, and except for cmp_B11, the docking affinities of the remaining candidate molecules surpass that of the original ligand of protein 4YL0 under the same docking conditions. Notably, candidate cmp_B14 demonstrated significantly improved docking affinity compared to the other candidates. Candidate cmp_B1 had hydrogen bond interactions with ArgB38 and AsnB46. Hydrogen bond interactions were established between Candidate cmp_B6 and AspB49. Candidate cmp_B7 generated hydrogen bond interactions with GluA77 and SerA127, while also forming a π-π stacking interaction with TyrA130. It is noteworthy that Candidate cmp_B7 has been reported to inhibit the generation of PGE_2_ [37]. Coupled with its hydrogen bond interactions with key amino acids at the active site of mPGES-1, these findings underscore the potential of cmp_B7 as an inhibitor for mPGES-1. Candidate cmp_B2 and cmp_B3 bound to the active pocket of protein 4YL1 with binding affinities of −9.19 and −8.82 kcal/mol, both of which were superior to the docking affinity of the original ligand of protein 4YL1 (−8.71 kcal/mol). Candidate cmp_B2 formed hydrogen bond interactions with AspB49 and ThrA131. Candidate cmp_B3 generated a hydrogen bond interaction with AspB49. Candidate cmp_B10 docked within the active domain of protein 5K0I, displaying a binding affinity of −10.31 kcal/mol, which outperformed the binding affinity of the original ligand of protein 5K0I under identical docking conditions (−8.57 kcal/mol). Candidate cmp_B10 engaged in hydrogen bond interactions with AspB49, SerA127, GluA77, and TyrA117. Candidates cmp_B8 and cmp_B9 bound within the active cavity of mPGES-1 (PDB: 5TL9), with binding affinities of −9.55 and −9.2 kcal/mol, respectively. The binding affinity of the original ligand within complex 5TL9 was −8.63 kcal/mol, indicating that cmp_B8 and cmp_B9 exhibited superior binding performances under the same docking parameters. Candidate cmp_B8 formed a hydrogen bond interaction with SerA127. Candidate cmp_B9 had hydrogen bond interactions with SerA127, GluA77, and TyrA117.

The majority of candidate molecules exhibited the ability to form hydrogen bond interactions with key amino acids of mPGES-1. Notably, the docking affinities of several candidate molecules even surpass those of the original ligands within the co-crystal complexes. These findings collectively contribute to bolstering the plausibility of the potential mPGES-1 inhibitors identified through our virtual screening process.

## 3. Materials and Methods

### 3.1. Construction of the Catalogue for MFH Substances

The culture of traditional Chinese medicine has a long history, the “Compendium of Materia Medica”, compiled in the 16th century, records more than 300 kinds of medicine and food homology (MFH) substances. In 2002, Chinese National Health Commission issued a list of “Chinese medicine that can be used in health food” (including 114 kinds of substances), both as traditional Chinese medicines and as food [49]. In 2018, Chinese National Health Commission published a list of 110 MFH substances [50]. Additionally, the 2020 edition of the “Pharmacopoeia” records 86 substances as “healthy food” and 193 substances as “therapeutic food”, both of which are in line with the concept of medicine and food homology [51]. Therefore, by examining and integrating the above four lists, a total of 503 MFH substances were collected in the catalogue of this study, the active ingredients were further collected based on this catalog with the aim of constructing a comprehensive database of MFH substances.

### 3.2. Collection and Preparation of Active Ingredients from MFH Substances

Based on the integrated and constructed catalogue of MFH substances, we collected the active ingredients of 503 MFH substances from the following public Chinese Medicine database: Traditional Chinese Medicines Integrated Database (TCMID) [19], and Traditional Chinese Medicine Systems Pharmacology Database and Analysis Platform (TCMSP) [20]. However, some of the MFH substances were not included in the above databases, or the collected active ingredients were inadequate. Therefore, we also conducted a literature search (92 pieces of Chinese literature from the China National Knowledge Infrastructure (CNKI) [52]) and manually collected the active ingredients of MFH substances to ensure the integrity and diversity of our MFH substances database. For the collected active ingredient molecules, we further disconnected group metals in simple salts, removed minor components, screened the duplicated molecules, and retrieved the isomeric SMILES of molecules to upgrade the data quality of our MFH substances database. As a result, we obtained 15,362 active ingredients of MFH substances from TCMID, 11,154 active ingredients from TCMSP, and an additional 803 active ingredients from 92 pieces of literature.

### 3.3. Chemical Space Analysis on the MFH Substances Database

Chemical space analysis is a widely utilized approach for exploring, comprehending, and optimizing a multitude of potential molecules [53]. Herein, we calculated molecular weight (MW) and octanol–water partition coefficient (LogP) on the collected active ingredient molecules. These two physicochemical properties enable us to measure and demonstrate the breadth of the chemical space distribution of the MFH database. MW and LogP were calculated via Python-based RDKit packages [54], and the chemical space distribution was visualized with Matplotlib [55]. For measuring the structural similarity of our MFH database, we computed the Tanimoto coefficients (TCs) [21] based on 1024 bits ECFP4 fingerprints by using RDKit packages [54]. To thoroughly analyze the diversity of molecular structures within the MFH database, we computed Murcko scaffolds of the molecules and conducted a clustering analysis using the K-means algorithm. Prior to clustering, we characterized the calculated Murcko scaffolds by ECFP4 fingerprints and employed T-distributed Stochastic Neighbor Embedding (t-SNE) [56] to reduce the high-dimensional data into two dimensions. ECFP4 fingerprints were computed via K-means clustering and t-SNE were implemented with scikit-learn [57].

### 3.4. Construction of Datasets for Building Classification and Regression Models

#### 3.4.1. Datasets for Modeling on COX-2 Inhibitors

Dataset 1 and 2 (Table 11) for developing classification models to classify highly/weakly active COX-2 inhibitors were identical to the datasets in our previous work [58]. Different thresholds were employed to label highly/weakly active inhibitors, which enabled the constructed classification models to cover different chemical spaces. This strategy would further contribute to the hit rate of VS on MFH substances. A total of 1511 molecules (Dataset 3) were collected from ChEMBL [59], Reaxys [60], and SciFinder [61], their IC_50_ values were tested by enzyme-linked immunoassay. The pIC_50_ (−log_10_IC_50_) values ranged from 5.06 to 9.52. Dataset 3 was utilized to develop regression models with the aim of accurately predicting the bioactivities of MFH substances. External validation sets A1 and A2 were collected from the newly published literature and used to evaluate the generalizability of the constructed classification and regression models, respectively. The IC_50_ values of molecules in the External validation set A2 were all tested by enzyme-linked immunoassay.

#### 3.4.2. Datasets for Modeling on mPGES-1 Inhibitors

Numerous mPGES-1 inhibitors with diverse structures were collected from ChEMBL, Reaxys, and SciFinder. Dataset 4 was composed of 3179 mPGES-1 inhibitors, their IC_50_ values vary from 0.0001 to 20,000 μM. Molecules with IC_50_ > 10 μM were weakly active inhibitors; with IC_50_ < 0.6 μM were highly active inhibitors. Dataset 5 comprised 3455 inhibitors with IC_50_ values ranging from 0.0001 to 20,000 μM. Molecules with IC_50_ ≥ 10 μM and IC_50_ < 10 μM are weakly and highly active inhibitors, respectively. Datasets 4 and 5 were employed to construct classification models on mPGES-1 inhibitors (shown in Table 9). Dataset 6 containing 735 inhibitors was derived from our previous work [62], pIC_50_ values of inhibitors in Dataset 6 ranged from 5.54 to 9 (all tested by homogeneous time-resolved fluorescence assay). External validation sets B1 and B2 were used to evaluate the generalizability of the constructed classification and regression models, respectively.

#### 3.4.3. Splitting Strategy for Generating the Training/Test Set

The datasets for modeling on COX-2 inhibitors were divided into training/test sets at the ratio of 4:1. The datasets for modeling on mPGES-1 inhibitors were randomly divided into training/test sets at the ratio of 3:1. The datasets of both COX-2 and mPGES-1 inhibitors were randomly split 10 times for generating the training/test sets to avoid the random error. The random splitting was conducted by using the function *StratifiedSplit* of the Python toolkit scikit-learn 0.22.1 [57].

### 3.5. Characterization of Datasets

#### 3.5.1. Binary Fingerprints for Classification Models

Three types of well-known fingerprints were employed to comprehensively and multifacetedly characterize the structural features of molecules within our MFH database. 166 bits MACCS fingerprints (belongs to dictionary-based fingerprints), 1024 bits Avalon fingerprints (topological-based fingerprints) [63], and 1024 bits ECFP4 fingerprints (circular fingerprints) [64] were computed with RDKit [54] packages. To avoid the inclusion of redundant information, the calculated fingerprints were then filtered by the variance, and fingerprints with variance in the bottom quartile were excluded from the construction of classification models.

#### 3.5.2. Physicochemical Molecular Descriptors for Regression Models

Two types of physicochemical molecular descriptors were utilized to represent molecules for further developing QSAR models. A total of 22 global molecular descriptors and 96 3D property-weighted autocorrelation from CORINA were calculated by the CORINA Symphony software V1.0 [65]. A total of 115 2D physicochemical molecular descriptors and 85 FragmentCount descriptors from RDKit were computed via MayaChemTools [66]. The calculated descriptors were further screened with Pearson correlation coefficient (PCC), and listed in descending order by recursive feature elimination with the random forest estimator (RF-RFE) before modeling; more details can be seen in the previous work [62]. Additionally, all the reserved calculated descriptors were auto-scaled to the same range from 0.1 to 0.9.

### 3.6. Supervised Machine Learning Algorithms for Modeling

Machine learning (ML) algorithms are capable of discerning relationships within large datasets and devising optimal approaches for their analysis without prior specification. Four supervised ML algorithms, including support vector machine (SVM) [67], random forest (RF) [68], deep neural networks (DNNs) [69], and eXtreme Gradient Boosting (XGBoost) [70], were utilized to provide predictions on COX-2 and mPGES-1 inhibitors.

#### 3.6.1. Modeling with SVM, RF, and XGBoost

The SVM algorithm equipped with radial basis function (RBF) kernel was involved in developing both classification and regression models, penalty parameter (*C*) and *γ* were hyperparameters to be confirmed during the optimization of the classification models. Except for the two parameters mentioned above, the insensitive parameter (*ε*) needed to be confirmed in the regression model’s optimization process. For RF models, the number of trees (*n_estimators*) and the maximum leaf nodes (*max_leaf_nodes*) were determined by the grid search. In modeling with XGBoost, the number of trees (*n_estimators*), the maximum depth of a tree (*max_depth*), the subsample ratio of the training instances (*subsample*), and the subsample ratio of columns when constructing each tree (*colsample_bytree*) were optimized by grid search. Other parameters not mentioned were set as their default values.

In addition to the common hyperparameters of SVM, RF, and XGBoost algorithms, the number of descriptors also served as a hyperparameter in the grid-based optimization of regression models. During the grid search, 5-fold cross-validation was repeated 10 times, and the optimal hyperparameters were determined based on the smallest mean squared error (MSE) of the 5-fold cross-validation [71]. Detailed ranges of those mentioned hyperparameters were listed in Appendix A.

#### 3.6.2. Modeling with DNN

Fully connected feed-forward neural networks with four hidden layers were constructed to develop classification and regression models. Neurons within the hidden layers were activated using the Relu function [72] and further compiled using the Adam optimizer with a learning rate of 0.0001. The training epoch was determined through repetitive 5-fold cross-validation training combined with early stopping [73]. For classification models, early stopping monitored changes in the predicted accuracy of the validation set. The validation set was generated through 5-fold cross-validation and was part of the training set. For regression models, early stopping focused on changes in the MSE values of the validation set. Training was halted when the accuracy or MSE of the validation set ceased to change within 50 epochs. To mitigate the potential for contingency arising from early stopping based on a single validation set, each cross-validation was conducted through repetitive training 50 times.

### 3.7. Ensemble Learning Based on Developed Classification Models

Ensemble learning integrates the predictions of multiple machine learning models to improve the robustness and accuracy of predictions made by individual base models [74]. In this study, the stacked generalization [75] was employed to combine the predicted probabilities of classification models built using the same algorithm but on datasets characterized by different fingerprints. These probabilities were then input into a logistic regression algorithm to generate predicted values of ensemble models with the same algorithm.

### 3.8. Unsupervised Machine Learning on MFH Substances

Unsupervised learning algorithms are trained on unlabeled data to discover hidden patterns or relationships among the data, as the hidden patterns and relationships can serve as a foundation for exploratory analysis [76]. The self-organized map (SOM) is a type of unsupervised artificial neural network; during the training of SOM, when the input layer receives data with similar vectors, these vectors are mapped onto the same neuron or neurons that are close together in the two-dimensional grid [77]. In this study, SOM was applied to perform clustering analysis on molecules of the MFH database, and further predict their inhibitory effects on COX-2 and mPGES-1.

### 3.9. Evaluation of Model Performances

The predicted accuracy (Q) and the Matthews correlation coefficient (MCC) were utilized as indicators of the performances of classification models. The coefficient of determination (R^2^) and root mean squared error (RMSE) were applied to evaluate the performances of regression models. These criteria mentioned above were calculated by the following equations:(1)$$Q=\frac{\mathrm{TP}+\mathrm{TN}}{\;\mathrm{TP}+\mathrm{TN}+\mathrm{FP}+\mathrm{FN}}\times \;100\%\;$$
(2)$$\mathrm{MCC}=\frac{\mathrm{TP}\times \mathrm{TN}-\mathrm{FP}\times \mathrm{FN}}{\sqrt{\left(\mathrm{TP}+\mathrm{FP}\right)\left(\mathrm{TP}+\mathrm{FN}\right)\left(\mathrm{TN}+\mathrm{FP}\right)\left(\mathrm{TN}+\mathrm{FN}\right)}}$$
where true positives (TPs) and true negatives (TNs) represent the number of “1” and “0” that were correctly predicted, respectively. False positives (FPs) and false negatives (FNs) represent the number of “1” and “0” that were wrongly predicted, respectively.
(3)$$R^{2}\left(y,\hat{y}\right)=1-\frac{\sum_{i=1}^{n}{\left(y_{i}-{\hat{y}}_{i}\right)}^{2}}{\sum_{i=1}^{n}{\left(y_{i}-\overset{-}{y}\right)}^{2}}$$
(4)$$\mathrm{RMSE}\left(y,\hat{y}\right)=\frac{1}{n}\sum_{i=1}^{n}{\left(y_{i}-{\hat{y}}_{i}\right)}^{2}$$
where n represents the total number of compounds; y represents an observed value of a compound; $\hat{y}$ represents predicted value of a compound; $\bar{y}$ represents the average of y.

### 3.10. Pan Assay Interference Compounds (PAINS) Screening

The pan assay interference compounds (PAINS) screening has evolved into a pivotal element within drug design. The PAINS rule is introduced to identify false positive compounds (frequent hitters) during biological screening initiatives. We obtained a list of known aggregators (12,645 molecules were shown in Appendix A) from Aggregator Advisor [78], which represented the known aggregator molecules and our screened MFH candidates with MACCS fingerprints. We employed RDKit to match the structure of each screened MFH candidate molecule with 12,645 known aggregator molecules. As a result, none of MFH candidates (10 potential COX-2 inhibitors and 15 potential mPGES-1 inhibitors) appeared in the known aggregators list.

### 3.11. Molecular Docking

In this study, molecular docking was conducted using the latest release of the widely used open-source program AutoDock Vina 1.2.0 [79]. Given the availability of multiple COX-2 and mPGES-1 crystal structures, the selection of an appropriate receptor with a low resolution is a prerequisite for reliable docking computations. Therefore, we chose COX-2 and mPGES-1 co-complex crystal structures from the PDB database that exhibited similar bound ligand to the screened medicine and food homologous (MFH) candidates. Ligands within the complex crystal structures and the screened MFH candidates were characterized using MACCS fingerprints. Subsequently, RDKit functions were employed to calculate Tanimoto similarity between the molecular structures of these entities. For each screened MFH candidate molecule, docking was performed with the crystal structure that exhibited the highest structural similarity (details were listed in Appendix A). Before formal docking, the ligands within the original complex crystal structures underwent re-docking. The better the alignment between the re-docked ligand and the experimentally determined ligand, the more optimal the parameter settings and system preparation of the docking calculation. Results of the re-docking of ligands within the complex crystal structures are presented in Appendix A. The protein preparation process involved the removal of water and other solvents, repair of missing residue sections, addition of hydrogen atoms to heavy atoms, and subsequent pre-docking energy minimization of the entire protein. The ligand preparation process included the addition of hydrogen atoms, computation of Gasteiger charges for all atoms, definition of rotatable bonds, and energy optimization. The grid box was adjusted based on the spatial center of the ligand within the crystal structure. Vina force field was employed during docking, with the exhaustiveness parameter set to 32.

## 4. Conclusions

In this study, we constructed a comprehensive database of 27,319 active ingredient molecules from 503 different types of medicine and food homologous (MFH). Analysis of the distribution of molecular weight (MW) and octanol–water partition coefficient (LogP) showed a wide range of values, indicating that our MFH database covers a wide chemical space. Structural diversity was assessed using Tanimoto coefficients (TCs), showing significant structural differences between molecules, with 95.92% of molecule pairs having TC values below 0.6. In addition, we performed Murcko scaffold analysis and K-means clustering, resulting in the identification of 11 different clusters in the MFH database. Among them, flavonoid clusters were the most abundant, followed by fatty acids, saponins, and sterols. The database was further enriched by the presence of lignans, alkaloids, triterpenoids, sesquiterpenoids, diterpenoids, and stilbenoids. Furthermore, we summarized the top 20 Murcko scaffolds, revealing diverse structures ranging from simple aromatic compounds with a single ring to complex systems with 7–8 rings. These findings collectively demonstrate the comprehensiveness and high structural diversity of our MFH database. Our MFH database will serve as a foundation for future studies, as it could facilitate the assessment of the effects of MFH on health and identifies potential mechanisms to accelerate the development of MFH-inspired products with nutritional and therapeutic value.

Based on datasets with different distributions of bioactivities, we employed four supervised learning algorithms (RF, SVM, DNN, and XGBoost), incorporating various fingerprints and physicochemical descriptors for modeling. As a result, a total of 240 classification models and 80 QSAR models were constructed for COX-2 and mPGES-1 inhibitors, respectively. Additionally, we also utilized ensemble learning to develop classification models. Based on the constructed single classifiers, another 80 integrated classification models were constructed for COX-2 and mPGES-1 inhibitors, respectively. For COX-2 inhibitors, ModelA_ensemble_RF_1, built on Dataset 1, demonstrated the best classification performance with MCC values of 0.802 and 0.603 on the test set and external validation set, respectively. ModelB_MACCS_SVM_6, constructed using Dataset 2, achieved the highest performance with MCC values of 0.657 and 0.572 on the test set and external validation set, respectively. ModelC_RDKIT_SVM_2, the optimal regression model based on Dataset 3, yielded RMSE values of 0.419 and 0.513 on the test set and external validation set, respectively. For mPGES-1 inhibitors, ModelD_ECFP_SVM_4 emerged as the top-performing classification model on Dataset 4, exhibiting MCC values of 0.832 and 0.584 on the test set and external validation set, respectively. ModelE_ECFP_SVM_1, the best classification model for Dataset 5, achieved MCC values of 0.799 and 0.579 on the test set and external validation set, respectively. ModelF_3D_SVM_1, based on Dataset 6, served as the optimal regression model with RMSE values of 0.253 and 0.35 on the test set and external validation set, respectively. These well-performing machine learning models can serve as powerful tools for virtual screening of the constructed MFH database, aiming to identify potential COX-2 and mPGES-1 inhibitors from MFH substances. Moreover, these models can be employed to predict the inhibitory capabilities of unknown compounds against COX-2 and mPGES-1, thus facilitating the discovery of novel anti-inflammatory drugs.

Finally, by means of a cascade ligand-based virtual screening strategy and a PAINS screening rule, we identified 10 potential COX-2 inhibitors and 15 potential mPGES-1 inhibitors from the MFH database. We verified candidates by molecular docking, investigated the interaction of the candidate molecules upon binding to COX-2 or mPGES-1. It is worth mentioning that some of these molecules have been previously reported to exhibit COX-2 inhibitory or anti-inflammatory activities. This demonstrates the effectiveness of the cascaded ligand-based virtual screening strategy employed in this study and provides design and modification ideas for the development of new effective anti-inflammatory drugs targeting COX-2 and mPGES-1.

## Figures and Tables

**Figure 1 molecules-28-06782-f001:**
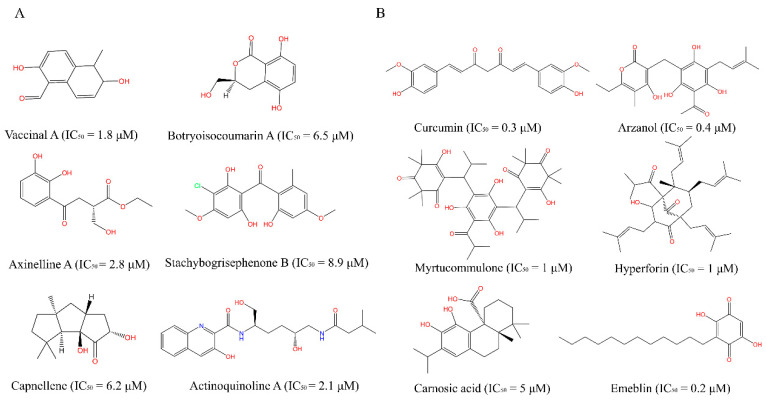
Active ingredients of natural products with anti-inflammatory effects. (**A**) Molecules with COX-2 inhibition; (**B**) Molecules with mPGES-1 inhibition.

**Figure 2 molecules-28-06782-f002:**
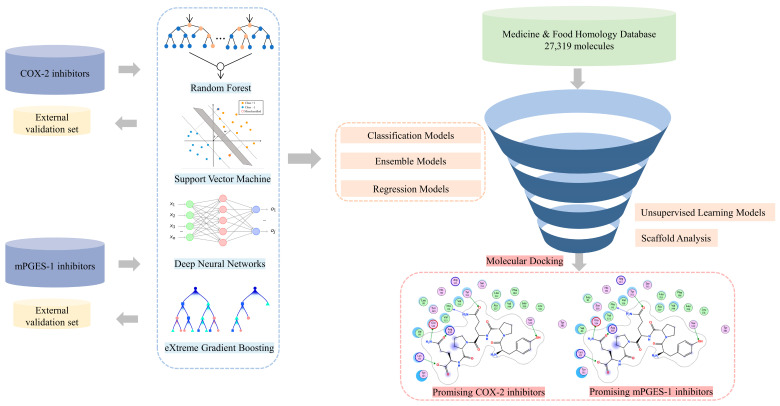
The workflow of this study. The pink circle in the interaction diagram represents polar amino acids, the green represents hydrophobic amino acids.

**Figure 3 molecules-28-06782-f003:**
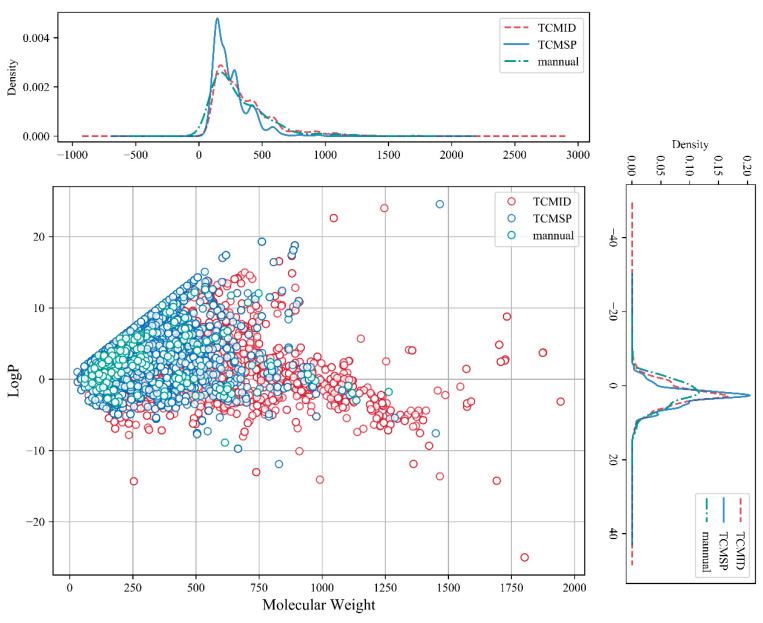
The distribution of molecular weight (MW) and LogP on the MFH substances database. The dark red line/dot, the navy-blue line/dot, and the green line/dot represent molecules derived from TCMID database, TCMSP database, and the literature, respectively.

**Figure 4 molecules-28-06782-f004:**
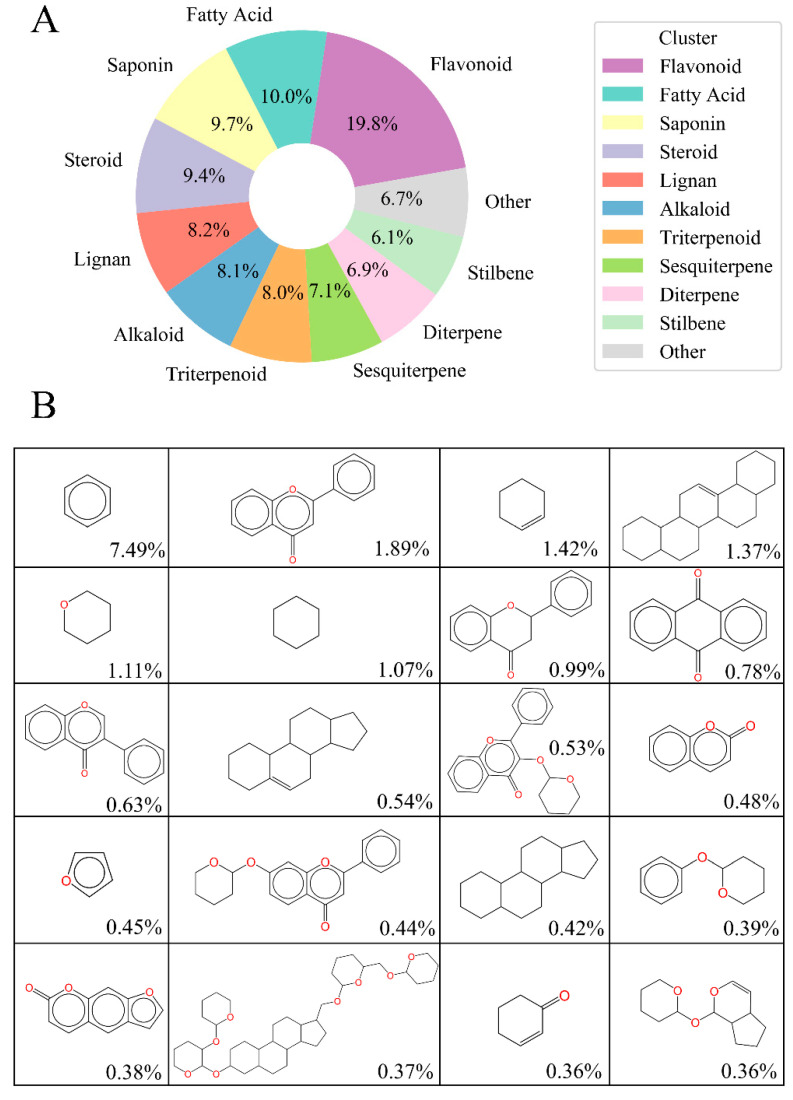
The overview of the chemical structures in the medicine and food homology (MFH) database. (**A**) The 11 classes of molecules of our MFH database after clustering analysis; (**B**) Top 20 Murcko scaffolds in the MFH database; percentages of molecules containing the Murcko scaffolds are labeled in the bottom-right corner.

**Figure 5 molecules-28-06782-f005:**
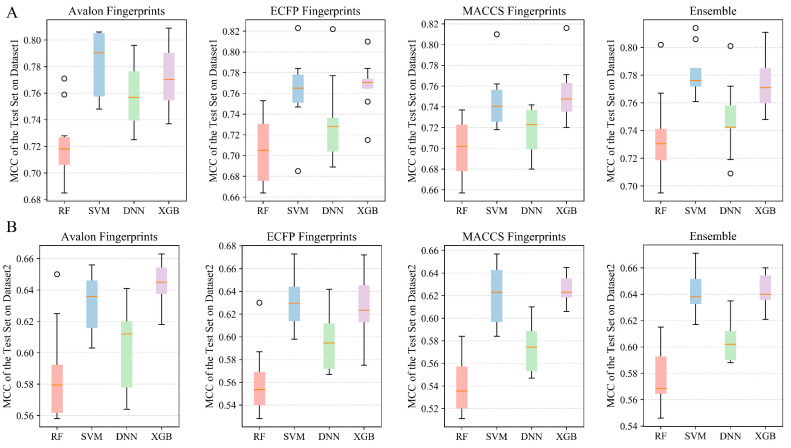
(**A**) MCC values of single classification models and ensemble models constructed using three fingerprints and four machine learning algorithms for the test set based on Dataset 1. (**B**) MCC values of single classification models and ensemble models constructed using three fingerprints and four machine learning algorithms for Dataset 2 for the test set. The solid red line in the middle of the box chart represents the median, and the circle represents outliers.

**Figure 6 molecules-28-06782-f006:**
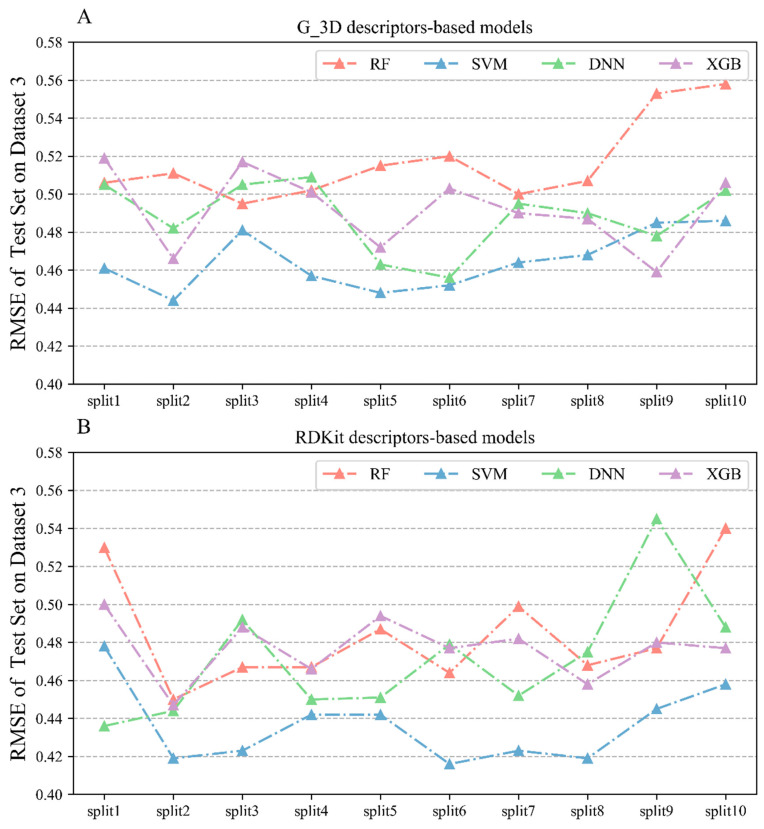
(**A**) RMSE values on the test sets of the regression models constructed using Corina 3D (represented as G_3D) on Dataset 3; (**B**) RMSE values on the test sets of the regression models constructed using RDKit descriptors for the test set based on Dataset 3.

**Figure 7 molecules-28-06782-f007:**
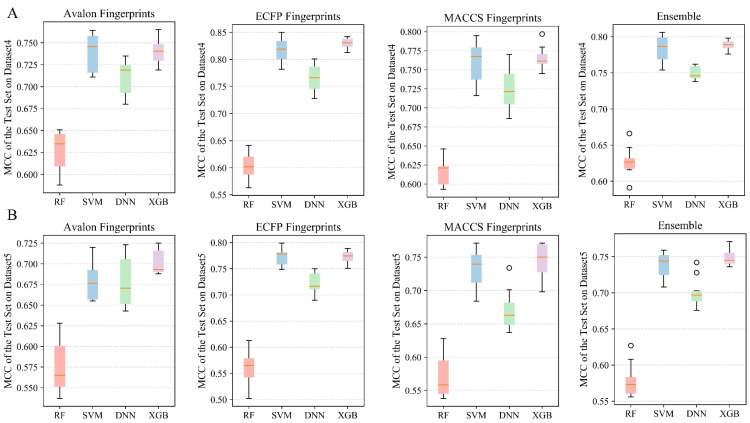
(**A**) MCC values of single classification models and ensemble models constructed using three fingerprints and four machine learning algorithms for Dataset 4 for the test set. (**B**) MCC values of single classification models and ensemble models constructed using three fingerprints and four machine learning algorithms for the test set based on Dataset 5. The solid red line in the middle of the box chart represents the median, and the circle represents outliers.

**Figure 8 molecules-28-06782-f008:**
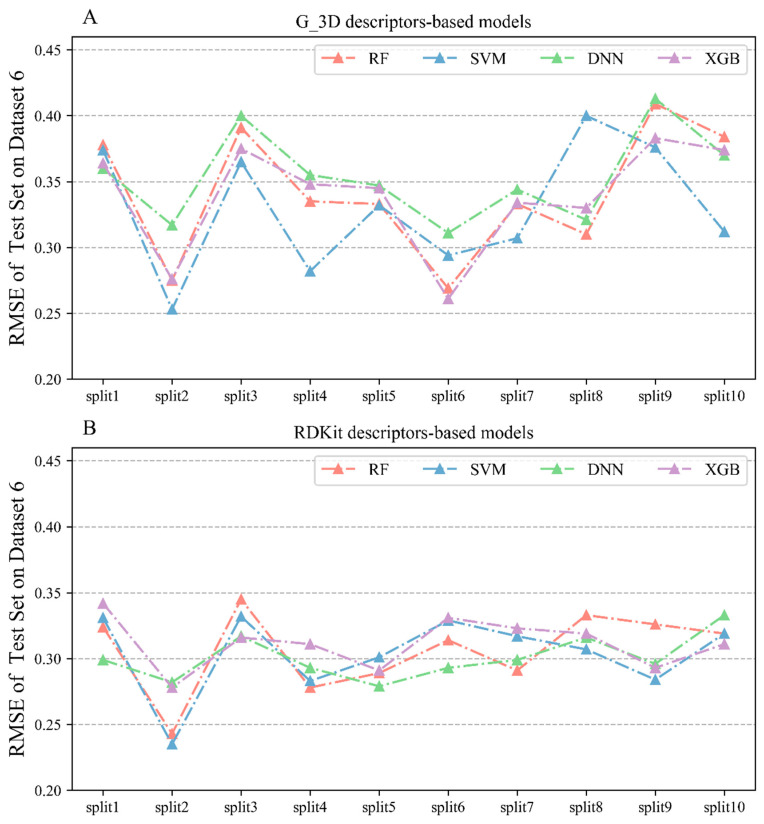
(**A**) RMSE values on the test sets of the regression models constructed using Corina 3D (represented as G_3D) on Dataset 6; (**B**) RMSE values on the test sets of the regression models constructed using RDKit descriptors for the test set based on Dataset 6.

**Figure 9 molecules-28-06782-f009:**
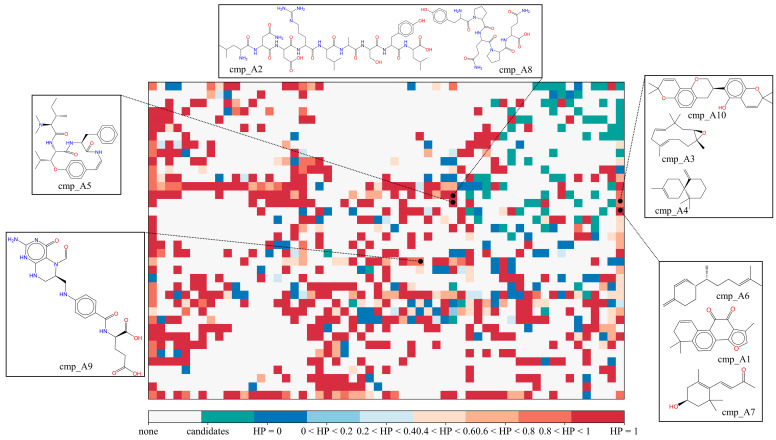
The mapping positions of COX-2 inhibitor candidates in self-organizing map. HP represents the proportion of the number of highly active inhibitors in the grid to all molecules in the grid. The larger the calculated HP of one grid (close to 1), the warmer the color tends to be (red). On the contrary, the closer the HP value of a grid is to 0, the cooler the color tends to be (blue).

**Figure 10 molecules-28-06782-f010:**
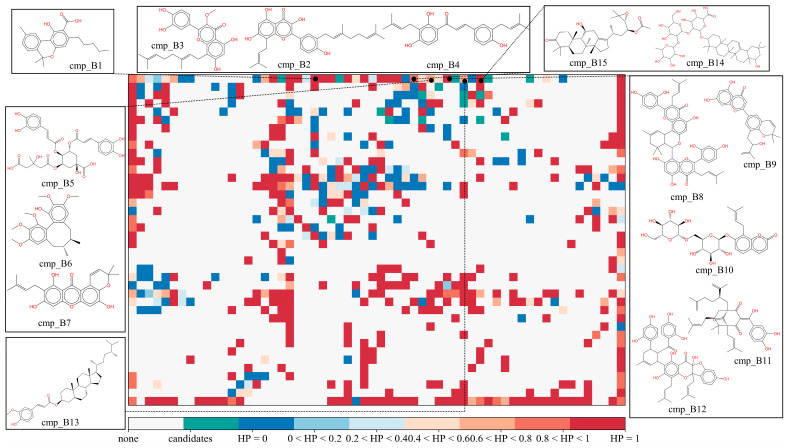
The mapping positions of mPGES-1 inhibitors candidates in self-organizing map. HP represents the proportion of the number of highly active inhibitors in the grid to all molecules in the grid. The larger the calculated HP of one grid (close to 1), the warmer the color tends to be (red). On the contrary, the closer the HP value of a grid is to 0, the cooler the color tends to be (blue).

**Table 1 molecules-28-06782-t001:** The overall performances of classification models on Dataset 1.

Fingerprints	Algorithm	Training Set		5-Fold Cross-Validation		Test Set		External Validation Set A1	
		Q (ave_std)	MCC (ave_std)	Q (ave_std)	MCC (ave_std)	Q (ave_std)	MCC (ave_std)	Q (ave_std)	MCC (ave_std)
Avalon	RF	0.911 ± 0.005	0.817 ± 0.009	0.853 ± 0.007	0.7 ± 0.015	0.863 ± 0.014	0.721 ± 0.026	0.777 ± 0.027	0.524 ± 0.008
Avalon	SVM	0.97 ± 0.007	0.939 ± 0.014	0.892 ± 0.006	0.779 ± 0.012	0.893 ± 0.011	0.783 ± 0.023	0.786 ± 0.037	0.531 ± 0.017
Avalon	DNN	0.996 ± 0.002	0.991 ± 0.005	0.972 ± 0.006	0.942 ± 0.012	0.883 ± 0.011	0.759 ± 0.024	0.778 ± 0.033	0.532 ± 0.011
Avalon	XGBoost	0.975 ± 0.024	0.949 ± 0.054	0.8893 ± 0.031	0.794 ± 0.061	0.89 ± 0.011	0.774 ± 0.023	0.792 ± 0.036	0.541 ± 0.02
ECFP4	RF	0.91 ± 0.008	0.814 ± 0.017	0.85 ± 0.005	0.691 ± 0.01	0.857 ± 0.015	0.704 ± 0.031	0.763 ± 0.02	0.508 ± 0.024
ECFP4	SVM	0.988 ± 0.007	0.975 ± 0.014	0.885 ± 0.007	0.764 ± 0.015	0.885 ± 0.016	0.763 ± 0.033	0.794 ± 0.023	0.537 ± 0.022
ECFP4	DNN	0.994 ± 0.001	0.988 ± 0.002	0.975 ± 0.005	0.948 ± 0.011	0.87 ± 0.019	0.732 ± 0.038	0.773 ± 0.021	0.515 ± 0.017
ECFP4	XGBoost	0.99 ± 0.003	0.981 ± 0.005	0.883 ± 0.002	0.761 ± 0.004	0.887 ± 0.011	0.768 ± 0.023	0.767 ± 0.02	0.531 ± 0.013
MACCS	RF	0.897 ± 0.004	0.79 ± 0.008	0.844 ± 0.005	0.68 ± 0.01	0.854 ± 0.012	0.701 ± 0.025	0.737 ± 0.018	0.519 ± 0.011
MACCS	SVM	0.945 ± 0.013	0.887 ± 0.027	0.874 ± 0.007	0.742 ± 0.014	0.876 ± 0.013	0.746 ± 0.026	0.758 ± 0.024	0.532 ± 0.011
MACCS	DNN	0.973 ± 0.005	0.945 ± 0.01	0.939 ± 0.009	0.874 ± 0.019	0.862 ± 0.012	0.716 ± 0.023	0.741 ± 0.015	0.526 ± 0.01
MACCS	XGBoost	0.968 ± 0.006	0.931 ± 0.014	0.875 ± 0.005	0.745 ± 0.011	0.881 ± 0.012	0.753 ± 0.025	0.775 ± 0.02	0.54 ± 0.023
Ensemble	RF	0.933 ± 0.018	0.835 ± 0.016	0.876 ± 0.021	0.717 ± 0.02	0.886 ± 0.024	0.735 ± 0.029	0.787 ± 0.025	0.535 ± 0.023
Ensemble	SVM	0.986 ± 0.007	0.952 ± 0.013	0.901 ± 0.006	0.78 ± 0.01	0.903 ± 0.009	0.782 ± 0.016	0.797 ± 0.017	0.551 ± 0.012
Ensemble	DNN	0.996 ± 0.002	0.986 ± 0.007	0.971 ± 0.006	0.931 ± 0.012	0.883 ± 0.014	0.747 ± 0.025	0.775 ± 0.013	0.535 ± 0.009
Ensemble	XGBoost	0.987 ± 0.008	0.963 ± 0.018	0.893 ± 0.009	0.777 ± 0.018	0.896 ± 0.009	0.774 ± 0.019	0.787 ± 0.017	0.546 ± 0.014

Dataset 1 contains 1630 COX-2 inhibitors (the number of inhibitors in the training/test set = 1304:326); external validation set A1 contains 368 COX-2 inhibitors. Q: Accuracy; MCC: Matthews Correlation Coefficient, ave_std: average values and standard deviation values, Ensemble: combined the predicted probabilities of classification models built with Avalon, ECFP4, and MACCS fingerprints.

**Table 2 molecules-28-06782-t002:** The overall performances of classification models on Dataset 2.

Fingerprints	Algorithm	Training Set		5-Fold Cross-Validation		Test Set		External Validation Set A1	
		Q (ave_std)	MCC (ave_std)	Q (ave_std)	MCC (ave_std)	Q (ave_std)	MCC (ave_std)	Q (ave_std)	MCC (ave_std)
Avalon	RF	0.838 ± 0.004	0.67 ± 0.008	0.786 ± 0.006	0.566 ± 0.012	0.795 ± 0.014	0.586 ± 0.029	0.747 ± 0.017	0.505 ± 0.01
Avalon	SVM	0.932 ± 0.017	0.863 ± 0.035	0.814 ± 0.006	0.625 ± 0.013	0.817 ± 0.008	0.632 ± 0.018	0.785 ± 0.014	0.529 ± 0.016
Avalon	DNN	0.992 ± 0.002	0.984 ± 0.003	0.943 ± 0.007	0.883 ± 0.015	0.802 ± 0.012	0.602 ± 0.025	0.747 ± 0.21	0.507 ± 0.01
Avalon	XGBoost	0.97 ± 0.019	0.938 ± 0.035	0.818 ± 0.007	0.628 ± 0.013	0.825 ± 0.006	0.644 ± 0.014	0.762 ± 0.026	0.523 ± 0.015
ECFP4	RF	0.828 ± 0.005	0.65 ± 0.011	0.778 ± 0.005	0.548 ± 0.011	0.784 ± 0.014	0.561 ± 0.028	0.713 ± 0.026	0.497 ± 0.013
ECFP4	SVM	0.948 ± 0.025	0.894 ± 0.052	0.811 ± 0.003	0.616 ± 0.006	0.817 ± 0.01	0.63 ± 0.022	0.757 ± 0.023	0.532 ± 0.018
ECFP4	DNN	0.989 ± 0.002	0.977 ± 0.003	0.956 ± 0.005	0.91 ± 0.01	0.8 ± 0.012	0.595 ± 0.024	0.725 ± 0.012	0.509 ± 0.009
ECFP4	XGBoost	0.947 ± 0.023	0.896 ± 0.043	0.811 ± 0.006	0.617 ± 0.012	0.813 ± 0.013	0.625 ± 0.028	0.755 ± 0.046	0.523 ± 0.022
MACCS	RF	0.811 ± 0.006	0.617 ± 0.012	0.77 ± 0.006	0.534 ± 0.013	0.773 ± 0.011	0.541 ± 0.022	0.713 ± 0.013	0.497 ± 0.009
MACCS	SVM	0.897 ± 0.016	0.792 ± 0.033	0.809 ± 0.004	0.614 ± 0.01	0.812 ± 0.012	0.621 ± 0.025	0.758 ± 0.027	0.529 ± 0.022
MACCS	DNN	0.946 ± 0.01	0.893 ± 0.017	0.887 ± 0.01	0.772 ± 0.02	0.79 ± 0.01	0.575 ± 0.022	0.732 ± 0.011	0.506 ± 0.008
MACCS	XGBoost	0.909 ± 0.012	0.816 ± 0.025	0.807 ± 0.004	0.611 ± 0.007	0.817 ± 0.009	0.626 ± 0.012	0.753 ± 0.013	0.515 ± 0.01
Ensemble	RF	0.84 ± 0.006	0.66 ± 0.008	0.792 ± 0.007	0.563 ± 0.011	0.798 ± 0.011	0.576 ± 0.02	0.739 ± 0.016	0.513 ± 0.01
Ensemble	SVM	0.94 ± 0.009	0.864 ± 0.019	0.827 ± 0.008	0.633 ± 0.011	0.83 ± 0.007	0.642 ± 0.016	0.78 ± 0.014	0.544 ± 0.011
Ensemble	DNN	0.989 ± 0.005	0.965 ± 0.007	0.942 ± 0.006	0.868 ± 0.012	0.812 ± 0.008	0.604 ± 0.015	0.748 ± 0.004	0.52 ± 0.003
Ensemble	XGBoost	0.953 ± 0.014	0.895 ± 0.027	0.823 ± 0.005	0.63 ± 0.008	0.83 ± 0.006	0.643 ± 0.012	0.767 ± 0.018	0.531 ± 0.01

Dataset 2 contains 2925 COX-2 inhibitors (the number of inhibitors in the training/test set = 2340:585); external validation set A1 contains 368 COX-2 inhibitors. Q: Accuracy; MCC: Matthews Correlation Coefficient, ave_std: average values and standard deviation values, Ensemble: combined the predicted probabilities of classification models built with Avalon, ECFP4, and MACCS fingerprints.

**Table 3 molecules-28-06782-t003:** The overall performances of regression models on Dataset 3.

Descriptors	Algorithm	Training Set		5-Fold Cross-Validation		Test Set		External Validation Set A2	
		R^2^ (ave_std)	RMSE (ave_std)	R^2^ (ave_std)	RMSE (ave_std)	R^2^ (ave_std)	RMSE (ave_std)	R^2^ (ave_std)	RMSE (ave_std)
G_3D	RF	0.847 ± 0.004	0.346 ± 0.004	0.633 ± 0.01	0.537 ± 0.003	0.651 ± 0.007	0.517 ± 0.021	0.539 ± 0.063	0.634 ± 0.036
G_3D	SVM	0.941 ± 0.011	0.214 ± 0.023	0.713 ± 0.007	0.475 ± 0.006	0.725 ± 0.017	0.465 ± 0.014	0.553 ± 0.05	0.612 ± 0.034
G_3D	DNN	0.951 ± 0.021	0.191 ± 0.041	0.833 ± 0.06	0.354 ± 0.062	0.691 ± 0.025	0.488 ± 0.018	0.536 ± 0.066	0.63 ± 0.036
G_3D	XGBoost	0.907 ± 0.022	0.268 ± 0.035	0.817 ± 0.024	0.375 ± 0.026	0.684 ± 0.031	0.492 ± 0.02	0.539 ± 0.05	0.634 ± 0.032
RDKit	RF	0.858 ± 0.007	0.334 ± 0.009	0.679 ± 0.008	0.503 ± 0.007	0.688 ± 0.025	0.485 ± 0.028	0.532 ± 0.041	0.607 ± 0.021
RDKit	SVM	0.959 ± 0.012	0.179 ± 0.027	0.744 ± 0.01	0.449 ± 0.01	0.746 ± 0.022	0.436 ± 0.019	0.581 ± 0.05	0.596 ± 0.038
RDKit	DNN	0.965 ± 0.042	0.143 ± 0.076	0.862 ± 0.096	0.307 ± 0.102	0.72 ± 0.026	0.471 ± 0.031	0.594 ± 0.037	0.591 ± 0.03
RDKit	XGBoost	0.988 ± 0.002	0.095 ± 0.01	0.923 ± 0.012	0.243 ± 0.021	0.703 ± 0.025	0.477 ± 0.015	0.594 ± 0.038	0.584 ± 0.035

Dataset 3 contains 1511 COX-2 inhibitors with IC_50_ values tested in vitro by enzyme-linked immunoassay (the number of inhibitors in the training/test set = 1209:302) external validation set A2 contains 114 COX-2 inhibitors, R^2^: the coefficient of determination; RMSE: root mean squared error, ave_std: average values and standard deviation values.

**Table 4 molecules-28-06782-t004:** The overall performances of classification models on Dataset 4.

Fingerprints	Algorithm	Training Set		5-Fold Cross-Validation		Test Set		External Validation Set B1	
		Q (ave_std)	MCC (ave_std)	Q (ave_std)	MCC (ave_std)	Q (ave_std)	MCC (ave_std)	Q (ave_std)	MCC (ave_std)
Avalon	RF	0.914 ± 0.003	0.722 ± 0.012	0.882 ± 0.004	0.605 ± 0.014	0.888 ± 0.006	0.627 ± 0.022	0.744 ± 0.013	0.5 ± 0.013
Avalon	SVM	0.978 ± 0.003	0.931 ± 0.01	0.916 ± 0.004	0.736 ± 0.013	0.917 ± 0.006	0.739 ± 0.021	0.76 ± 0.006	0.513 ± 0.009
Avalon	DNN	0.992 ± 0.001	0.977 ± 0.005	0.966 ± 0.006	0.894 ± 0.019	0.908 ± 0.006	0.711 ± 0.019	0.747 ± 0.01	0.501 ± 0.011
Avalon	XGBoost	0.99 ± 0.004	0.971 ± 0.011	0.913 ± 0.004	0.73 ± 0.008	0.918 ± 0.005	0.74 ± 0.014	0.747 ± 0.017	0.508 ± 0.006
ECFP4	RF	0.912 ± 0.003	0.713 ± 0.012	0.878 ± 0.004	0.587 ± 0.016	0.882 ± 0.006	0.603 ± 0.024	0.726 ± 0.01	0.501 ± 0.013
ECFP4	SVM	0.993 ± 0.004	0.977 ± 0.012	0.94 ± 0.003	0.812 ± 0.01	0.941 ± 0.007	0.817 ± 0.022	0.77 ± 0.012	0.543 ± 0.02
ECFP4	DNN	0.999 ± 0.001	0.998 ± 0.001	0.985 ± 0.003	0.954 ± 0.008	0.925 ± 0.008	0.767 ± 0.024	0.768 ± 0.012	0.533 ± 0.01
ECFP4	XGBoost	0.995 ± 0.004	0.987 ± 0.01	0.933 ± 0.004	0.796 ± 0.008	0.946 ± 0.006	0.83 ± 0.009	0.778 ± 0.005	0.547 ± 0.015
MACCS	RF	0.907 ± 0.005	0.698 ± 0.019	0.88 ± 0.003	0.596 ± 0.011	0.885 ± 0.005	0.616 ± 0.017	0.704 ± 0.012	0.504 ± 0.01
MACCS	SVM	0.97 ± 0.008	0.906 ± 0.025	0.921 ± 0.003	0.75 ± 0.011	0.924 ± 0.008	0.76 ± 0.025	0.752 ± 0.016	0.536 ± 0.011
MACCS	DNN	0.985 ± 0.002	0.955 ± 0.007	0.959 ± 0.004	0.872 ± 0.012	0.912 ± 0.009	0.726 ± 0.028	0.729 ± 0.008	0.529 ± 0.012
MACCS	XGBoost	0.977 ± 0.005	0.933 ± 0.013	0.92 ± 0.005	0.749 ± 0.011	0.925 ± 0.005	0.765 ± 0.014	0.757 ± 0.017	0.536 ± 0.014
Ensemble	RF	0.923 ± 0.005	0.723 ± 0.011	0.892 ± 0.005	0.608 ± 0.01	0.897 ± 0.007	0.627 ± 0.019	0.736 ± 0.011	0.514 ± 0.01
Ensemble	SVM	0.99 ± 0.004	0.949 ± 0.014	0.936 ± 0.004	0.777 ± 0.007	0.938 ± 0.006	0.783 ± 0.017	0.772 ± 0.01	0.542 ± 0.011
Ensemble	DNN	0.997 ± 0.001	0.985 ± 0.008	0.977 ± 0.013	0.905 ± 0.041	0.927 ± 0.003	0.75 ± 0.008	0.76 ± 0.004	0.533 ± 0.005
Ensemble	XGBoost	0.997 ± 0.001	0.974 ± 0.005	0.932 ± 0.002	0.769 ± 0.005	0.939 ± 0.004	0.788 ± 0.006	0.771 ± 0.008	0.54 ± 0.008

Dataset 4 contains 3179 mPGES-1 inhibitors (the number of inhibitors in the training/test set = 2384:795); external validation set B1 contains 217 mPGES-1 inhibitors. Q: the coefficient of determination; MCC: root mean squared error, ave_std: average values and standard deviation values, Ensemble: combined the predicted probabilities of classification models built with Avalon, ECFP4, and MACCS fingerprints.

**Table 5 molecules-28-06782-t005:** The overall performances of classification models on Dataset 5.

Fingerprints	Algorithm	Training Set		5-Fold Cross-Validation		Test Set		External Validation Set B1	
		Q (ave_std)	MCC (ave_std)	Q (ave_std)	MCC (ave_std)	Q (ave_std)	MCC (ave_std)	Q (ave_std)	MCC (ave_std)
Avalon	RF	0.913 ± 0.003	0.694 ± 0.013	0.885 ± 0.003	0.583 ± 0.014	0.882 ± 0.008	0.575 ± 0.032	0.699 ± 0.012	0.495 ± 0.007
Avalon	SVM	0.967 ± 0.01	0.892 ± 0.035	0.911 ± 0.003	0.703 ± 0.013	0.904 ± 0.006	0.679 ± 0.022	0.712 ± 0.008	0.507 ± 0.007
Avalon	DNN	0.99 ± 0.007	0.968 ± 0.022	0.965 ± 0.007	0.884 ± 0.021	0.902 ± 0.009	0.678 ± 0.03	0.718 ± 0.009	0.509 ± 0.01
Avalon	XGBoost	0.987 ± 0.005	0.962 ± 0.008	0.911 ± 0.005	0.708 ± 0.013	0.908 ± 0.006	0.702 ± 0.014	0.713 ± 0.012	0.511 ± 0.007
ECFP4	RF	0.904 ± 0.004	0.658 ± 0.015	0.878 ± 0.003	0.552 ± 0.011	0.879 ± 0.008	0.559 ± 0.033	0.718 ± 0.005	0.495 ± 0.007
ECFP4	SVM	0.99 ± 0.005	0.969 ± 0.018	0.934 ± 0.003	0.78 ± 0.009	0.932 ± 0.004	0.773 ± 0.016	0.767 ± 0.011	0.538 ± 0.016
ECFP4	DNN	0.998 ± 0.002	0.994 ± 0.007	0.986 ± 0.005	0.954 ± 0.016	0.917 ± 0.005	0.722 ± 0.019	0.728 ± 0.008	0.511 ± 0.008
ECFP4	XGBoost	0.994 ± 0.003	0.981 ± 0.01	0.933 ± 0.003	0.769 ± 0.011	0.934 ± 0.004	0.772 ± 0.012	0.766 ± 0.01	0.537 ± 0.009
MACCS	RF	0.9 ± 0.003	0.646 ± 0.013	0.876 ± 0.003	0.543 ± 0.013	0.882 ± 0.007	0.571 ± 0.03	0.697 ± 0.007	0.489 ± 0.008
MACCS	SVM	0.969 ± 0.006	0.896 ± 0.019	0.92 ± 0.004	0.731 ± 0.012	0.921 ± 0.008	0.733 ± 0.028	0.736 ± 0.017	0.512 ± 0.006
MACCS	DNN	0.983 ± 0.002	0.945 ± 0.006	0.959 ± 0.003	0.861 ± 0.011	0.9 ± 0.01	0.67 ± 0.028	0.71 ± 0.009	0.498 ± 0.008
MACCS	XGBoost	0.976 ± 0.004	0.923 ± 0.013	0.918 ± 0.005	0.727 ± 0.01	0.924 ± 0.008	0.745 ± 0.025	0.737 ± 0.012	0.517 ± 0.011
Ensemble	RF	0.916 ± 0.003	0.676 ± 0.01	0.89 ± 0.002	0.569 ± 0.009	0.891 ± 0.005	0.578 ± 0.022	0.715 ± 0.005	0.504 ± 0.004
Ensemble	SVM	0.986 ± 0.005	0.929 ± 0.017	0.932 ± 0.002	0.748 ± 0.007	0.929 ± 0.004	0.738 ± 0.017	0.748 ± 0.008	0.529 ± 0.007
Ensemble	DNN	0.998 ± 0.002	0.979 ± 0.008	0.98 ± 0.004	0.91 ± 0.014	0.917 ± 0.006	0.7 ± 0.02	0.729 ± 0.005	0.516 ± 0.007
Ensemble	XGBoost	0.994 ± 0.004	0.964 ± 0.008	0.929 ± 0.004	0.744 ± 0.01	0.931 ± 0.004	0.749 ± 0.011	0.748 ± 0.01	0.531 ± 0.009

Dataset 5 contains 3455 mPGES-1 inhibitors (the number of inhibitors in the training/test set = 2591:864); external validation set B1 contains 217 mPGES-1 inhibitors. Q: the coefficient of determination; MCC: root mean squared error, ave_std: average values and standard deviation values, Ensemble: combined the predicted probabilities of classification models built with Avalon, ECFP4, and MACCS fingerprints.

**Table 6 molecules-28-06782-t006:** The overall performances of regression models on Dataset 6.

Descriptors	Algorithm	Training Set		5-Fold Cross-Validation		Test Set		External Validation Set B2	
		R^2^ (ave_std)	RMSE (ave_std)	R^2^ (ave_std)	RMSE (ave_std)	R^2^ (ave_std)	RMSE (ave_std)	R^2^ (ave_std)	RMSE (ave_std)
G_3D	RF	0.929 ± 0.023	0.175 ± 0.027	0.718 ± 0.21	0.351 ± 0.016	0.731 ± 0.052	0.342 ± 0.046	0.615 ± 0.052	0.46 ± 0.047
G_3D	SVM	0.864 ± 0.041	0.241 ± 0.037	0.737 ± 0.019	0.339 ± 0.016	0.75 ± 0.048	0.329 ± 0.045	0.634 ± 0.048	0.444 ± 0.052
G_3D	DNN	0.927 ± 0.029	0.174 ± 0.03	0.745 ± 0.01	0.331 ± 0.008	0.727 ± 0.042	0.354 ± 0.032	0.611 ± 0.042	0.472 ± 0.034
G_3D	XGBoost	0.92 ± 0.035	0.183 ± 0.037	0.746 ± 0.028	0.332 ± 0.022	0.741 ± 0.044	0.339 ± 0.039	0.625 ± 0.044	0.455 ± 0.045
RDKit	RF	0.925 ± 0.023	0.181 ± 0.026	0.745 ± 0.015	0.335 ± 0.009	0.776 ± 0.051	0.306 ± 0.029	0.619 ± 0.016	0.449 ± 0.037
RDKit	SVM	0.874 ± 0.035	0.232 ± 0.033	0.751 ± 0.019	0.33 ± 0.014	0.778 ± 0.038	0.304 ± 0.029	0.63 ± 0.023	0.424 ± 0.019
RDKit	DNN	0.939 ± 0.025	0.161 ± 0.033	0.834 ± 0.038	0.265 ± 0.035	0.787 ± 0.016	0.301 ± 0.016	0.625 ± 0.009	0.451 ± 0.031
RDKit	XGBoost	0.98 ± 0.018	0.079 ± 0.048	0.75 ± 0.017	0.331 ± 0.011	0.773 ± 0.028	0.311 ± 0.018	0.625 ± 0.012	0.445 ± 0.024

Dataset 6 contains 735 mPGES-1 inhibitors with IC_50_ values tested in vitro by homogeneous time-resolved fluorescence assay; the number of inhibitors in the training/test set of Dataset 6 = 551:184), external validation set B2 contains 60 mPGES-1 inhibitors, R^2^: the coefficient of determination; RMSE: root mean squared error, ave_std: average values and standard deviation values.

**Table 8 molecules-28-06782-t008:** The mPGES-1 inhibitors candidates from the MFH database by virtual screening.

Candidates	Origins	IC_50_ μM ^a^	SOM HP ^b^	Effects
cmp_B1	*Cannabis Sativa* L.	0.88	1	COX inhibition [30]
cmp_B2	*Ramulus Mori*	0.25	0.93	
cmp_B3	*Amomum longiligularg*	0.34	0.93	Anti-breast cancer [31]
cmp_B4	*Glycyrrhiza glabra* L.	0.80	0.93	PTP1B [32] and nitric oxide (NO) inhibition [33]
cmp_B5	*Gardeniae Fructus*	0.18	0.82	LOX inhibition [34]
cmp_B6	*Schisandra chinensis*	0.37	0.82	UDP-glucuronosyltransferase [35], and oxLDL inhibition [36]
cmp_B7	*Rhizoma Dioscoreae*	0.55	0.82	prostaglandin E_2_ reduction [37]
cmp_B8	*Ramulus Mori*	0.18	1	
cmp_B9	*Mori Cortex*	0.19	1	Tyrosinase inhibition [38]
cmp_B10	*Glehniae Radix*	0.20	1	
cmp_B11	*Colla*	0.37	1	Anti-inflammation [39]
cmp_B12	*Epimrdii Herba*	0.50	1	Anti-inflammation [40]
cmp_B13	*Coicis Semen*	0.27	1	Anti-inflammation [41]
cmp_B14	*Radix Puerariae*	0.20	1	Anti-oxidation [42], andenteritis treatment [43]
cmp_B15	*Alisma Orientale*	0.29	1	COX-2 inhibition [44]

^a^: the average predicted IC_50_ values by a series of optimal regression models. ^b^: the percentage of highly active inhibitors mapped to the target molecules in the same position during the self-organized map (SOM) training. It can also be taken as the probability of being predicted by an unsupervised algorithm as a highly active mPGES-1 inhibitor.

**Table 9 molecules-28-06782-t009:** The interactions between potential COX-2 inhibitors and COX-2 protein obtained by docking computations.

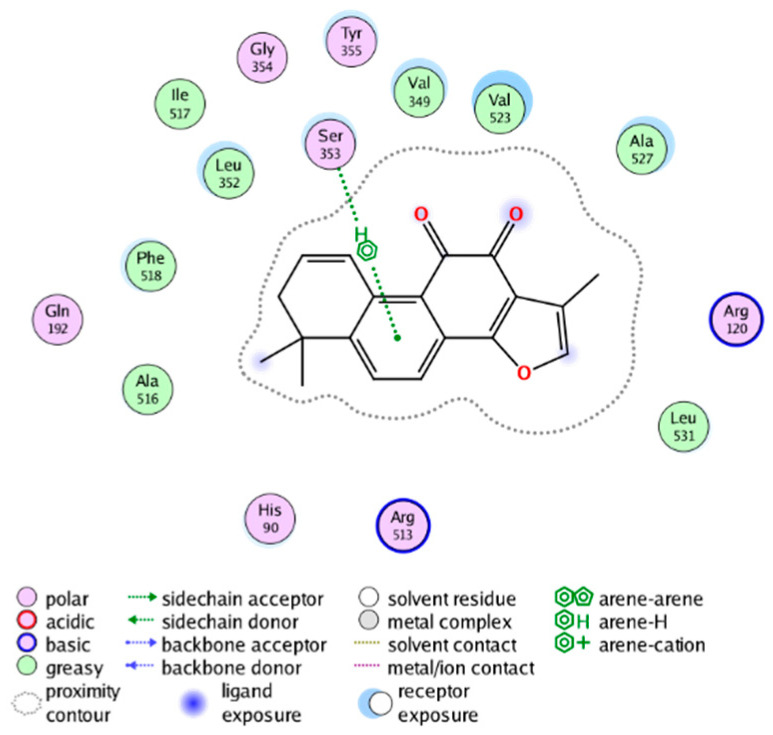	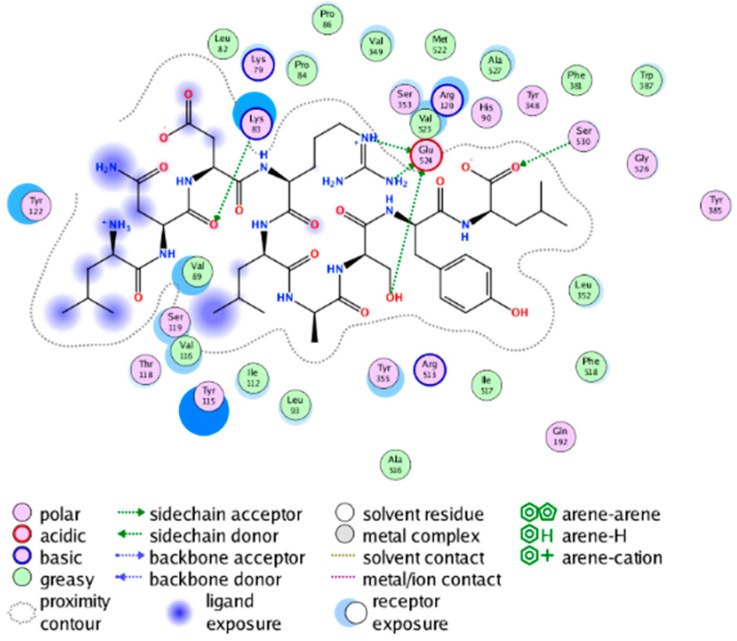
cmp_A1 & 5KIR ^a^ (affinity = −7.57 kcal/mol)	cmp_A2 & 6BL3 (affinity = −11.75 kcal/mol)
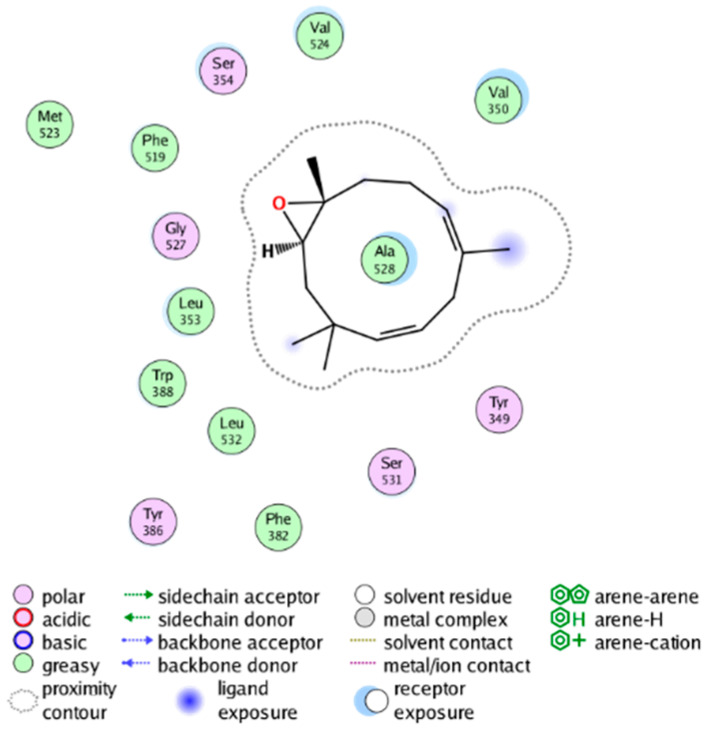	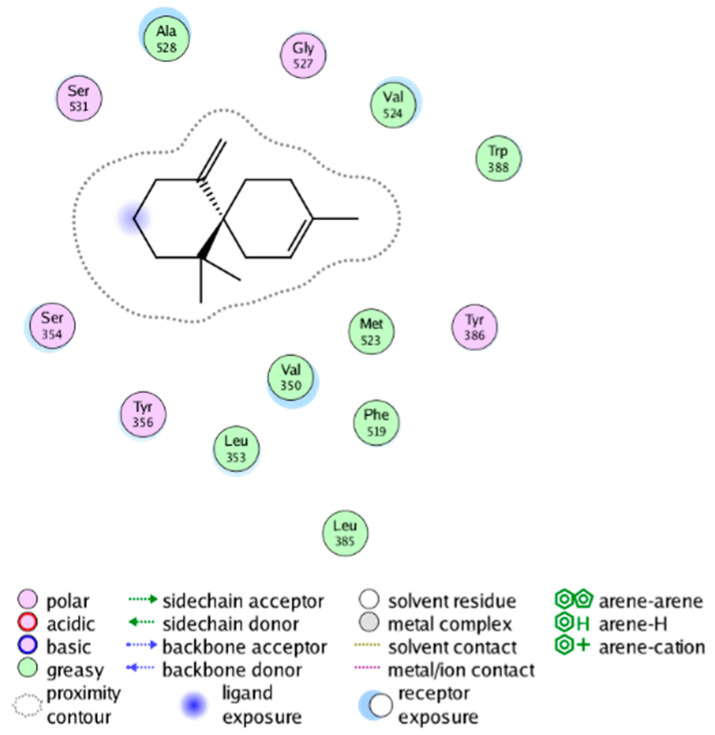
cmp_A3 & 4PH9 (affinity = −7.37 kcal/mol)	cmp_A4 & 4PH9 (affinity = −8.53 kcal/mol)
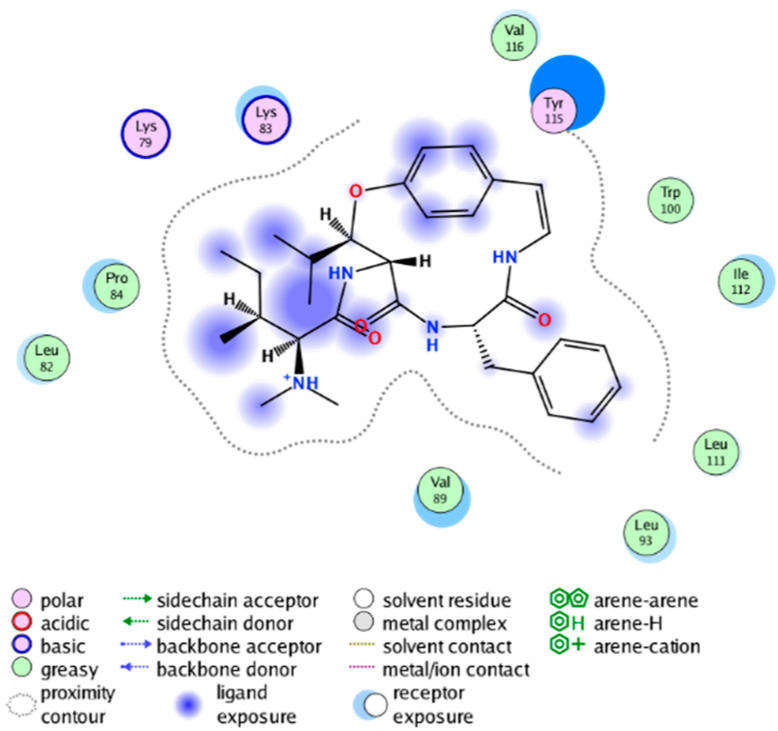	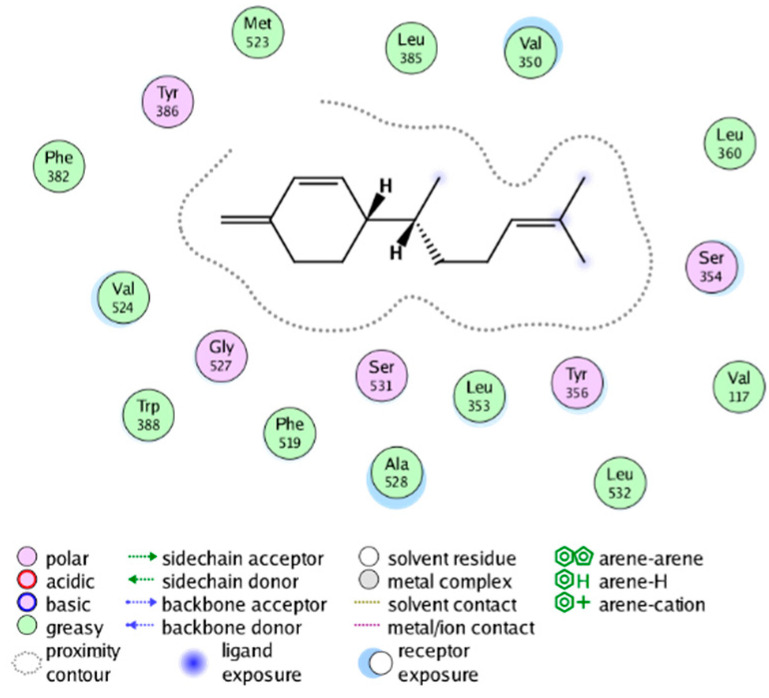
cmp_A5 & 6BL4 (affinity= −7.25 kcal/mol)	cmp_A6 & 4PH9 (affinity = −8.68 kcal/mol)
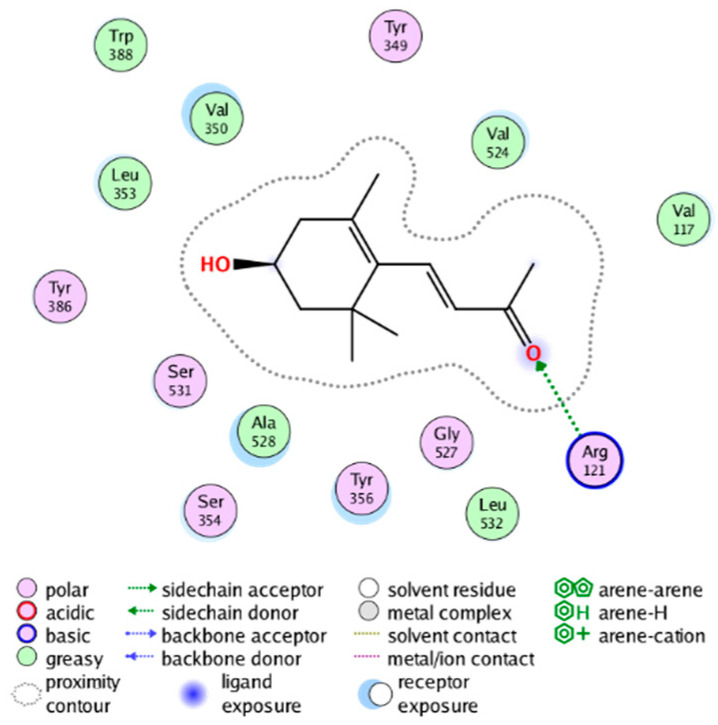	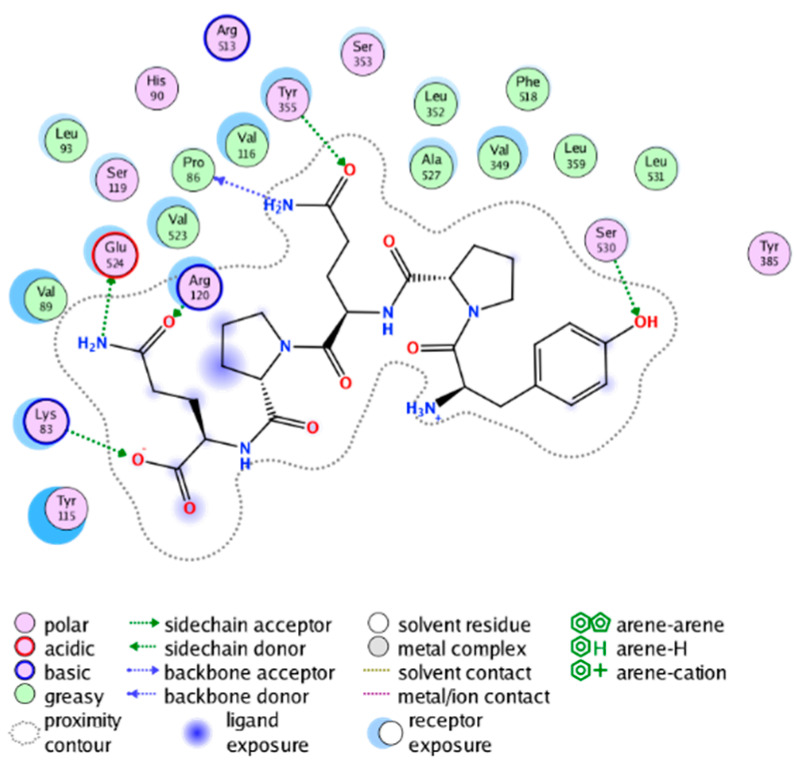
cmp_A7 & 4PH9 (affinity = −7.69 kcal/mol)	cmp_A8 & 6BL4 (affinity = −11.24 kcal/mol)
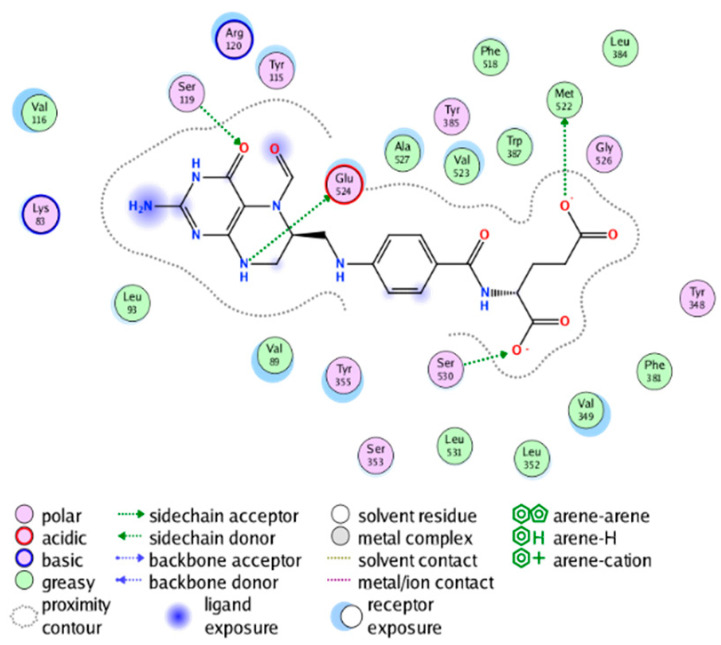	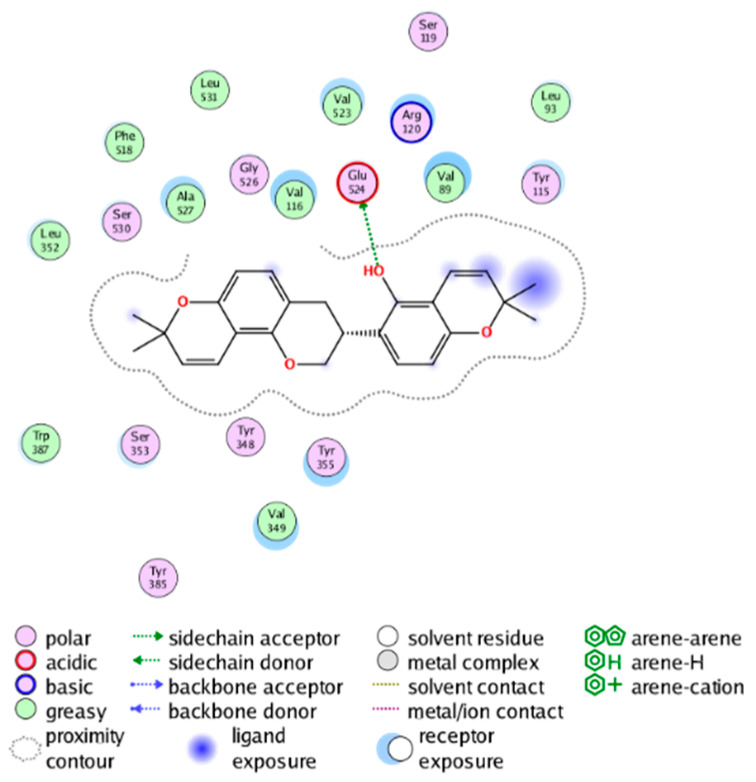
cmp_A9 & 6BL4 (affinity = −9.61 kcal/mol)	cmp_A10 & 6BL3 (affinity = −8.92 kcal/mol)

a: The PDB index of the receptor protein during docking.

**Table 10 molecules-28-06782-t010:** The interactions between potential mPGES-1 inhibitors and mPGES-1 protein obtained by molecular docking.

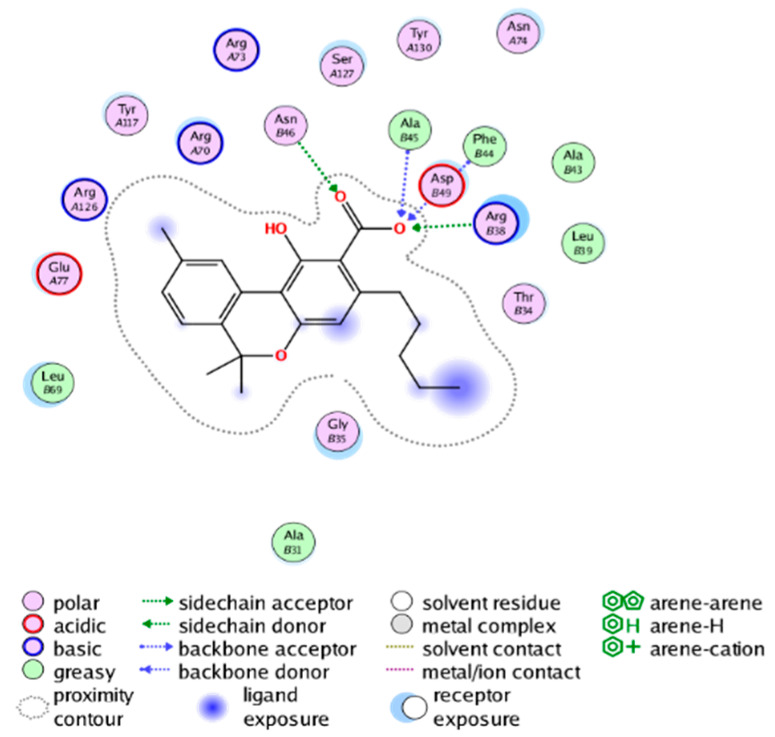	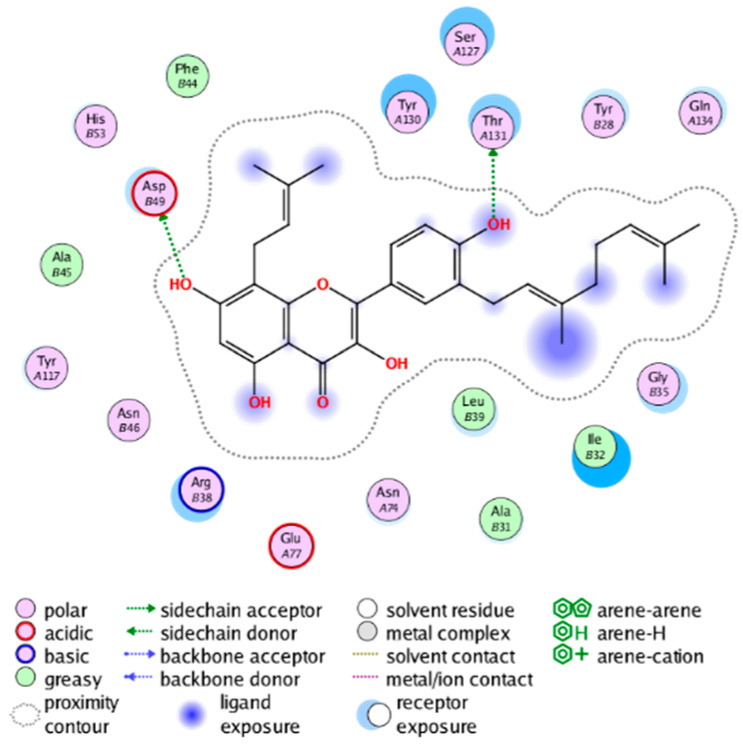
cmp_B1 & 4YL0 ^a^ (affinity = −7.93 kcal/mol)	cmp_B2 & 4YL1 (affinity = −9.19 kcal/mol)
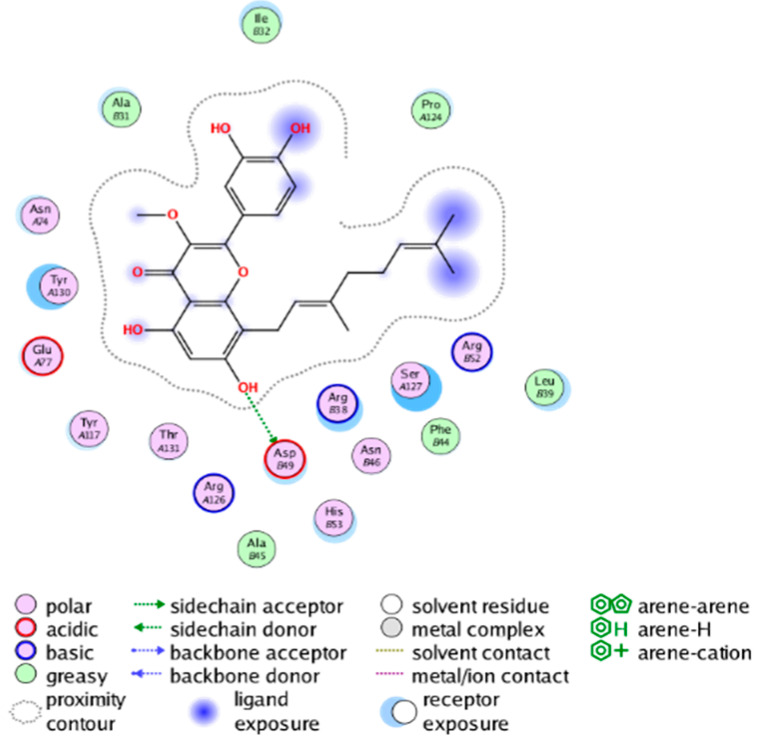	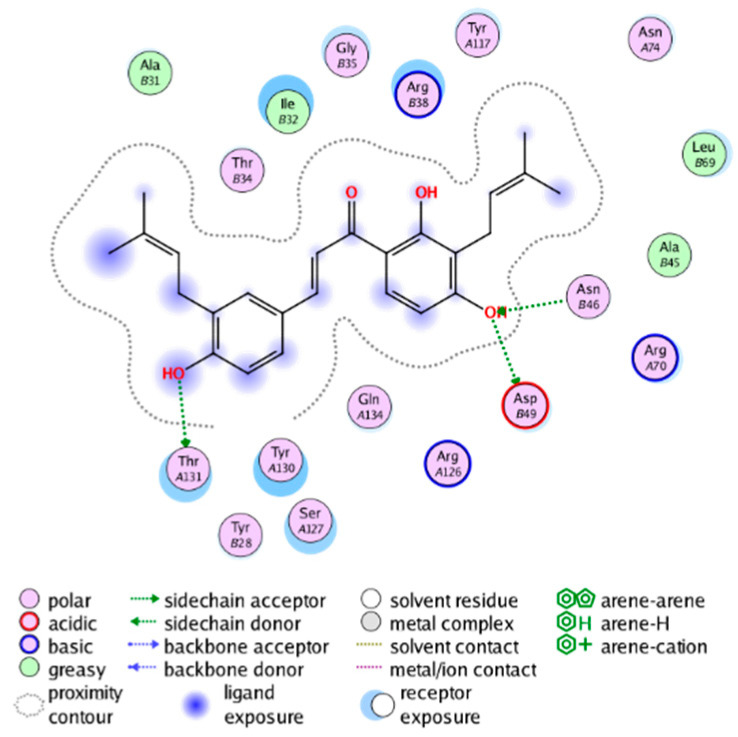
cmp_B3 & 4YL1 (affinity = −8.82 kcal/mol)	cmp_B4 & 4AL1 (affinity = −8.08 kcal/mol)
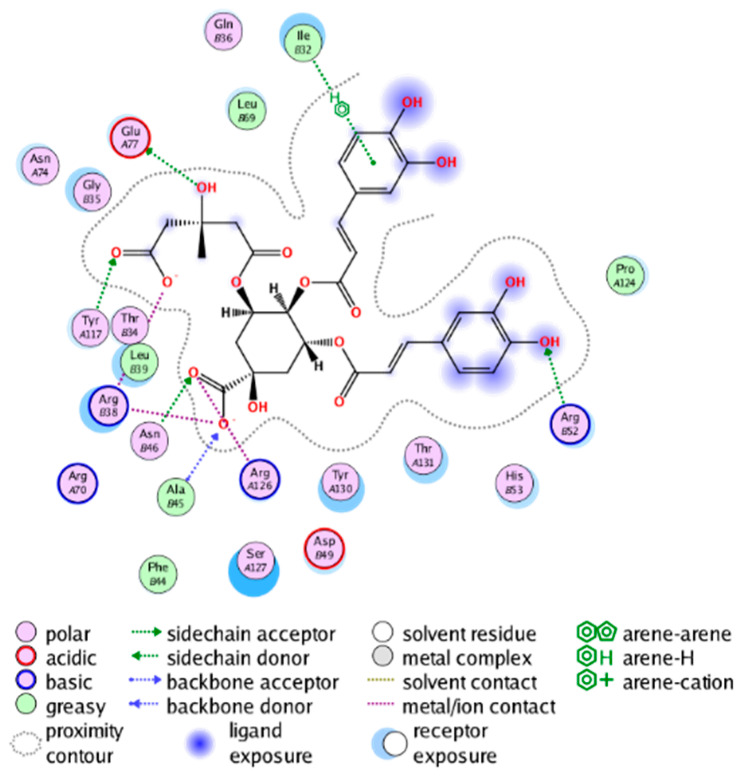	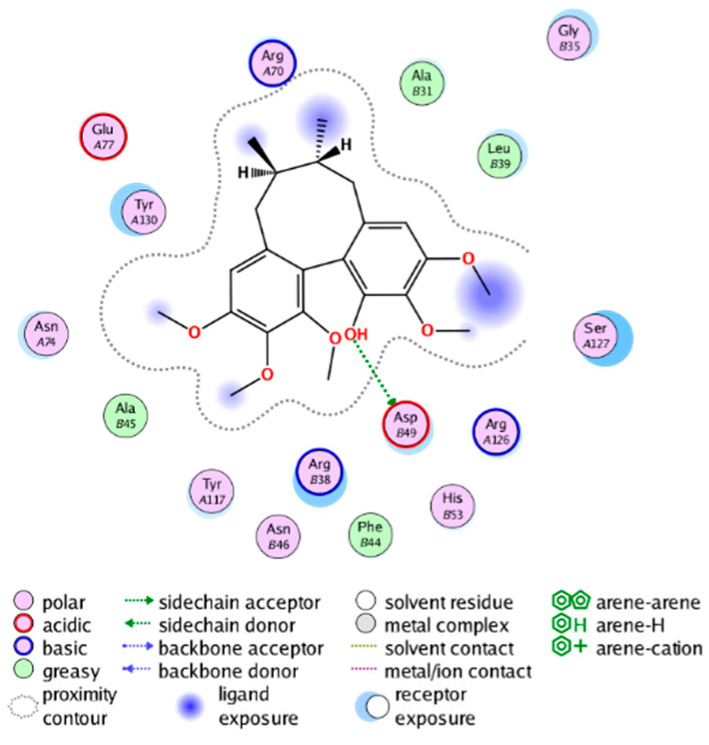
cmp_B5 & 4AL1 (affinity= −11.25 kcal/mol)	cmp_B6 & 4YL0 (affinity = −7.25 kcal/mol)
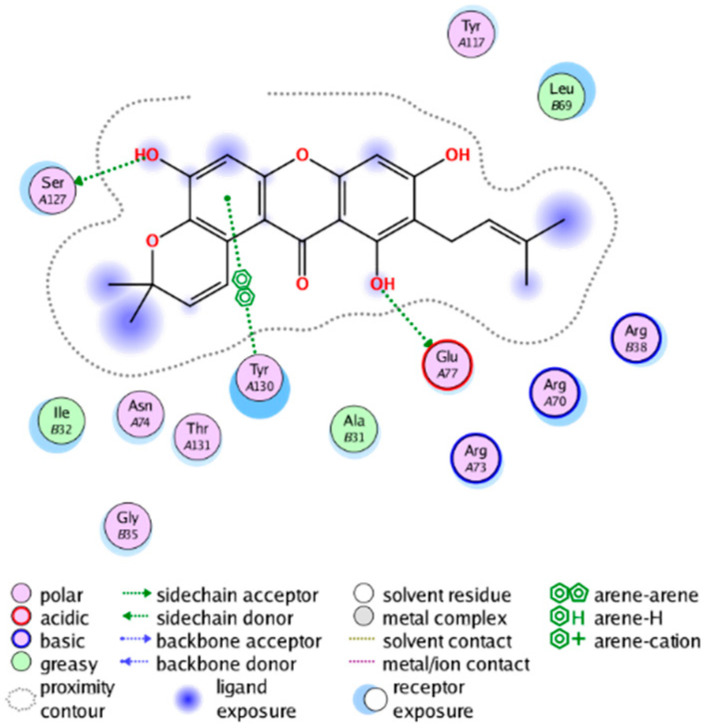	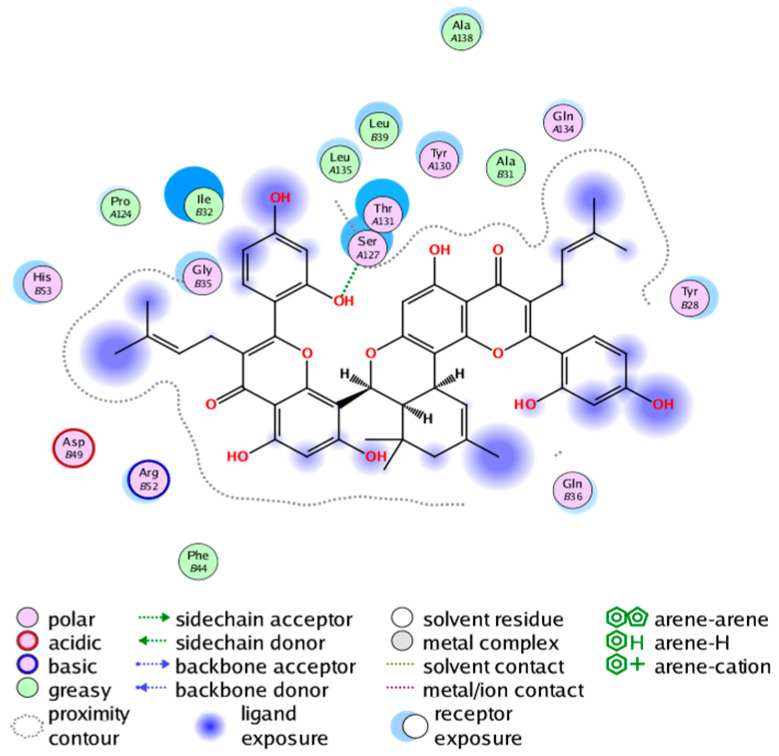
cmp_B7 & 4YL0 (affinity = −7.26 kcal/mol)	cmp_B8 & 5TL9 (affinity = −9.55 kcal/mol)
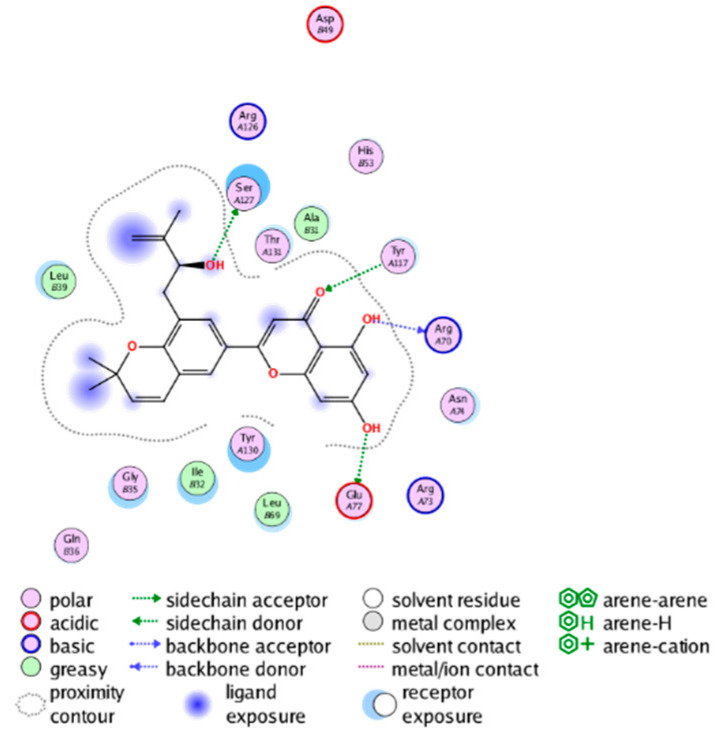	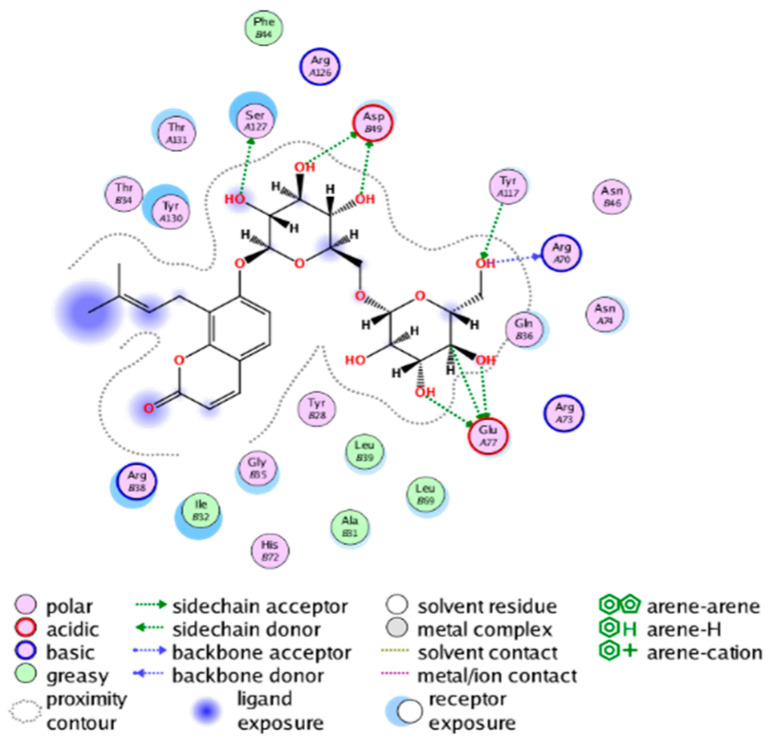
cmp_B9 & 5TL9 (affinity = −9.2 kcal/mol)	cmp_B10 & 5K0I (affinity = −10.31 kcal/mol)
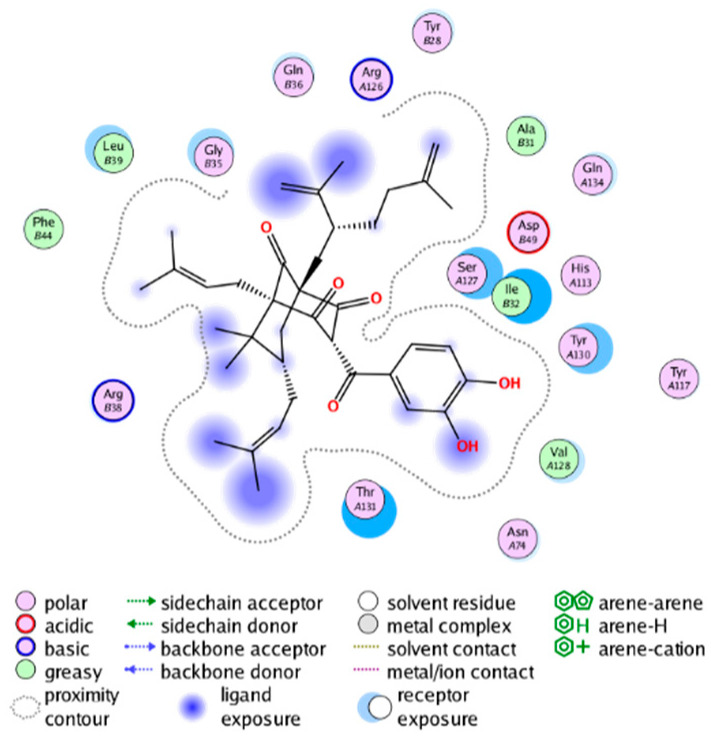	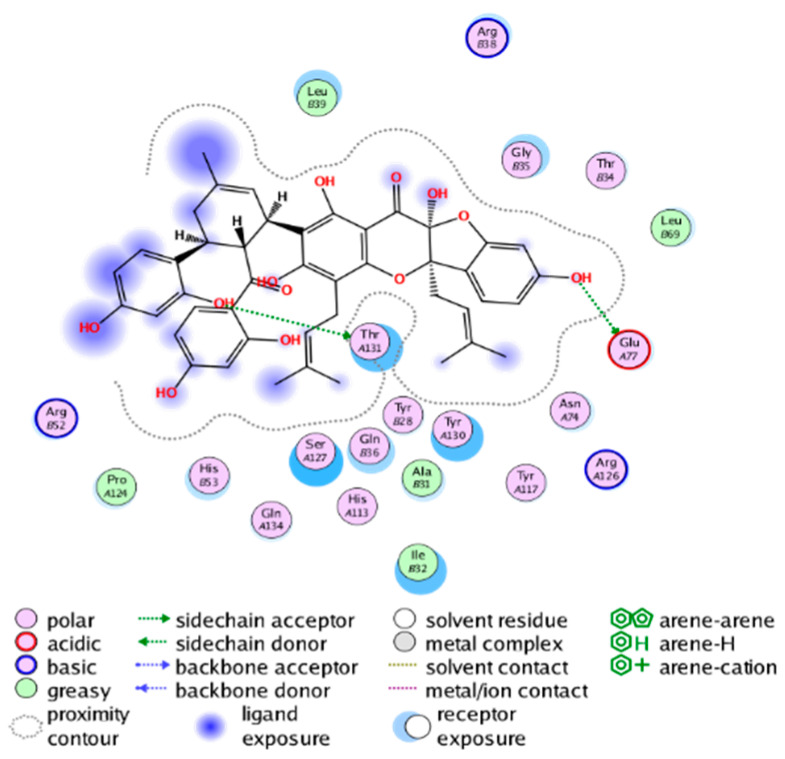
cmp_B11 & 4YL0 (affinity = −6.89 kcal/mol)	cmp_B12 & 4YL0 (affinity = −8.73 kcal/mol)
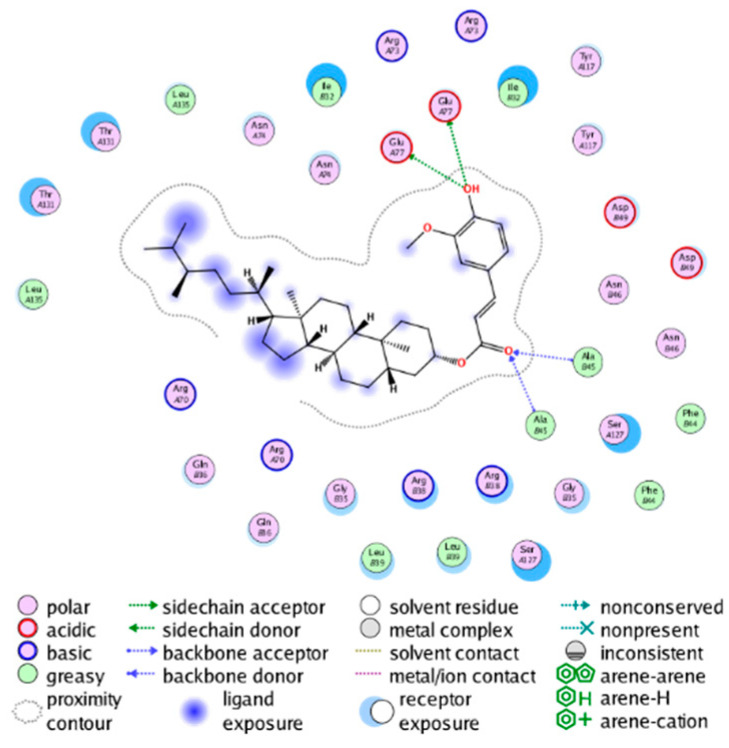	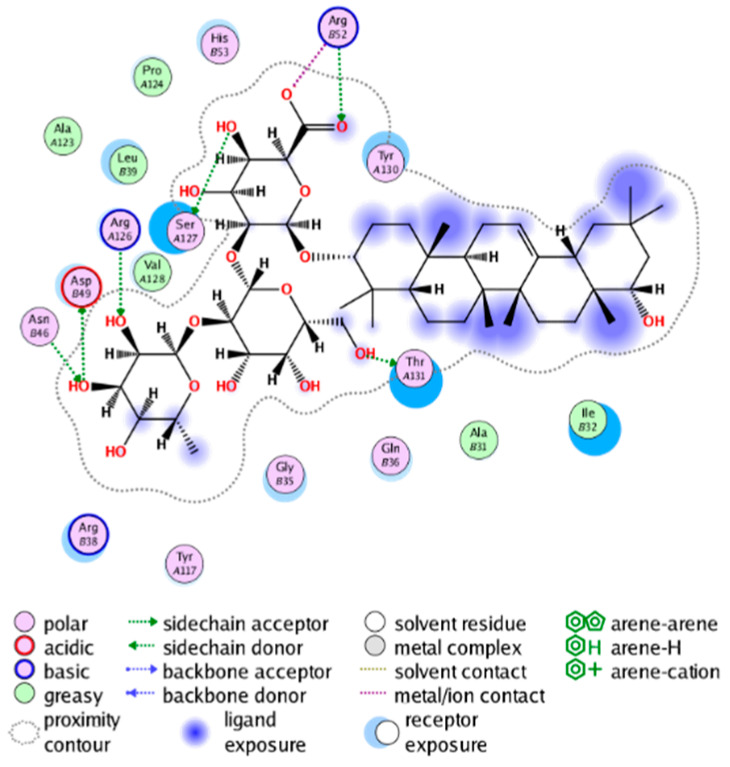
cmp_B13 & 4YL0 (affinity = −7.19 kcal/mol)	cmp_B14 & 4YL0 (affinity = −10.71 kcal/mol)
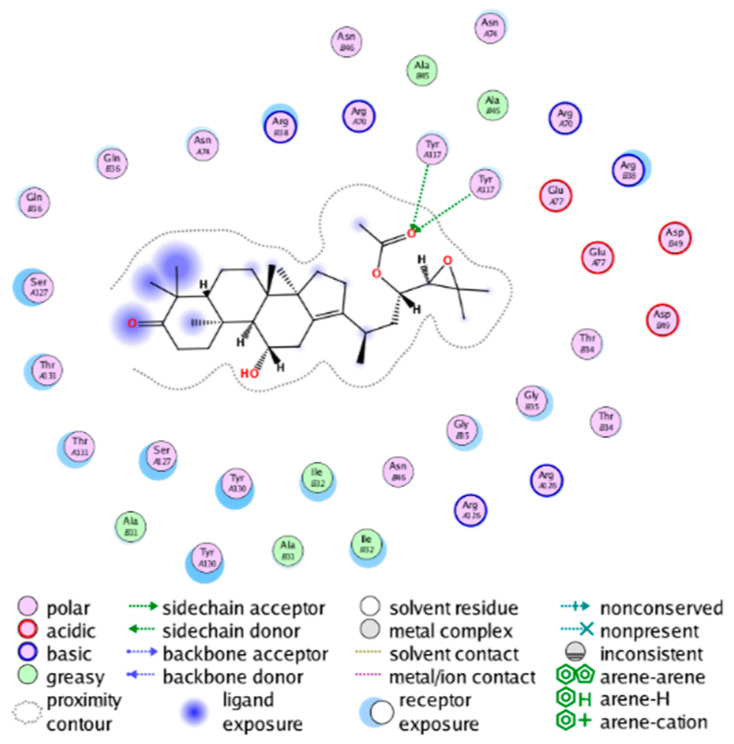	
cmp_B15 & 4YL0 (affinity = −7.34 kcal/mol)	

a: The PDB index of the receptor protein during docking.

**Table 11 molecules-28-06782-t011:** Datasets used in this study.

Datasets	Targets	N	Descriptions
Dataset 1	COX-2	1630	COX-2 inhibitors for constructing classification models, model results are shown in Table 1 and Appendix A, molecules with IC_50_ > 10 μM are weakly active inhibitors; with IC_50_ < 0.1 μM are highly active inhibitors
Dataset 2	COX-2	2925	COX-2 inhibitors for constructing classification models, model results are shown in Table 2 and Appendix A, molecules with IC_50_ > 1 μM are weakly active inhibitors; with IC_50_ ≤ 1 μM are highly active inhibitors
Dataset 3	COX-2	1511	COX-2 inhibitors for constructing QSAR models, model results are shown in Table 3 and Appendix A, molecules with IC_50_ values which were tested in vitro by enzyme-linked immunoassay
External validation set A1	COX-2	368	for evaluating the constructed classification models on COX-2 inhibitors
External validation set A2	COX-2	114	for evaluating the constructed regression models on COX-2 inhibitors
Dataset 4	mPGES-1	3179	mPGES-1 inhibitors for building classification models, model results are shown in Table 4 and Appendix A, molecules with IC_50_ > 10 μM are weakly active inhibitors; with IC_50_ < 0.6 μM are highly active inhibitors
Dataset 5	mPGES-1	3455	mPGES-1 inhibitors for building classification models, model results are shown in Table 5 and Appendix A, molecules with IC_50_ ≥ 10 μM are weakly active inhibitors; with IC_50_ < 10 μM are highly active inhibitors
Dataset 6	mPGES-1	735	mPGES-1 inhibitors for constructing QSAR models, model results are shown in Table 6 and Appendix A, molecules with IC_50_ values which were tested in vitro by homogeneous time-resolved fluorescence assay
External validation set B1	mPGES-1	217	for evaluating the constructed classification models on mPGES-1 inhibitors
External validation set B2	mPGES-1	60	for evaluating the constructed regression models on mPGES-1 inhibitors

N: the number of molecules in the datasets, COX-2: cyclooxygenase-2; mPGES-1: microsomal prostaglandin E2 synthase.

## Data Availability

All the codes used in this work can be found at: tyj-19951029/medicine-and-food-homologous-virtual-screnning (github.com, accessed on 3 June 2023). The MFH database established in this study can be obtained by sending an email to the author, and commercial use should be avoided.

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
