# Peer review of "Discovering the Active Ingredients of Medicine and Food Homologous Substances for Inhibiting the Cyclooxygenase-2 Metabolic Pathway by Machine Learning Algorithms"

_molecules, 2023, doi:10.3390/molecules28196782_

Round 1
Reviewer 1 Report
The manuscript by Tian et al. reports a machine learning approach for the identification of inhibitors of COX-2 and related pathways. The topic is of broad interest and the machine learning part itself is sound. However, the authors do not experimentally validate their predictions, which is an issue when stating that the 'study provides promising leads for the development of novel and effective anti-inflammatory therapies...'. Due to the hypothetical character of the results, I can support publication with limited enthusiasm.
The following issues should be addressed:
1) Either provide experimental data for the predicted bioactivity, or tone down the statements. In this regard the listing of predicted IC50 values is completely speculative (table 7 and 8). Since crystal structures of COX2 exists, docking of the proposed hits would be a way to support the findings.
2) The authors should carefully check weather the identified molecules qualify as leads. Some of them are very reactive, containing epoxide (A3), aldehydes (A9), or Michael acceptor scaffolds (A1, A7). Other molecules are quite large (A2, A8), which might not bind to the catalytic site of COX2. A discussion of these issues would be neccessary.
3) Do the authors have expected the outcome of the performance measures? Is there a rational explanation why rf-models perform worse compared to the others? Might the reason lay in the datasets used?
Author Response
Response to Reviewer 1 Comments
The manuscript by Tian et al. reports a machine learning approach for the identification of inhibitors of COX-2 and related pathways. The topic is of broad interest and the machine learning part itself is sound. However, the authors do not experimentally validate their predictions, which is an issue when stating that the 'study provides promising leads for the development of novel and effective anti-inflammatory therapies...'. Due to the hypothetical character of the results, I can support publication with limited enthusiasm.
The following issues should be addressed:
Point 1: Either provide experimental data for the predicted bioactivity, or tone down the statements. In this regard the listing of predicted IC50 values is completely speculative (table 7 and 8). Since crystal structures of COX2 exists, docking of the proposed hits would be a way to support the findings.
Response 1:
We deeply appreciate your thoughtful review of our manuscript. Your feedback has been invaluable in improving the quality and rigor of our work. We have carefully considered your comments and suggestions and have made the necessary revisions to address them. Below, we provide our responses to your specific concerns:
- Experimental data and speculative nature of predicted IC50 values:
We acknowledge the concern regarding the speculative nature of predicted IC50 values presented in Table 7 and 8. Considering we do not possess experimental data for the predicted bioactivity at this stage, we have emphasized that the IC50 values in the original manuscript are calculated by the machine learning model constructed in this work. Additionally, we have reframed our statements to accurately reflect the uncertainty of predicted IC50 values in the revised manuscript (highlighted in red in the Section of Conclusions).
- Supporting the findings through docking with COX-2 and mPGES-1crystal structures:
We thank you for suggesting the utilization of docking to support our findings, we believe this is a reasonable suggestion and have further performed docking using available COX-2 and mPGES-1crystal structures. The methods and results of molecular docking have been added in the revised manuscript (highlighted in red). Details are as follows:
“2.3.3 Molecular docking on the potential COX-2 and mPGES-1 inhibitors
Through a series of ligand-based virtual screening processes, we have identified potential inhibitors of COX-2 and mPGES-1 from the established MFH database, we further filtered these candidate molecules by the pan assay interference compounds (PAINS) rule [45] . To further validate the reliability and validity of the virtual screening, we conducted molecular docking (a widely utilized structure-based virtual screening approach) on the candidate MFH molecules. This procedure aimed to examine the binding modes between the candidate molecules and the proteins, while also assessing the ability of the candidate molecules to interact with the key amino acids of the target protein.”
“2.3.3.1 Molecular docking analysis on potential COX-2 inhibitors
The active site of COX-2 is demarcated from the initial substrate binding site by a constriction formed by three residues: Arg120, Tyr355, and Glu524. This structural constriction necessitates dilation to enable the entry or exit of substrates to and from the active site [46]. Ser530 and Tyr385 are also vital during the catalytic process of COX-2. Several typical binding modes of non-steroidal anti-inflammatory drugs (NSAIDs) interacting with COX-2 have been reported. These modes include the hydrophobic region of NSAIDs interacting with Tyr385, Trp387, and neighboring residues; the polar region of NSAIDs interacting with residues located above Tyr355; NSAIDs with negative charges interacting with Arg120 [47].
We calculated the structural similarity (measured by Tanimoto coefficient) between the candidate MFH molecules and the ligand molecules in the published crystal structures, we selected the protein crystal structure with the bound ligand having the highest structural similarity to the candidate MFH molecules for docking, details are shown in Table S10. As shown in Table 9, candidate compounds cmp_A3, cmp_A4, cmp_A6, and cmp_A7 bind within the active cavity of COX-2 (PDB: 4PH9), with binding affinities of -7.37, -8.53, -8.68, and -7.69 kcal/mol, respectively. The co-crystallized ligand, ibuprofen, in the complex 4PH9, under the same docking conditions, exhibits a binding affinity of -9.41 kcal/mol. While the four candidate molecules displayed affinities lower than the original ligand within the protein, they all established interactions with key amino acid residues in the catalytic domain of COX-2: hydrophobic interactions with Trp388 and polar interactions involving Tyr356, Ser531, and Tyr386. Candidate cmp_A7 generated a hydrogen bond interaction with Arg121, indicating a robust binding force. Candidate cmp_A1 bound within the active pocket of COX-2 (PDB: 5KIR), exhibiting a binding affinity of -7.57 kcal/mol. Meanwhile, the co-crystallized ligand within complex 5KIR achieved a binding affinity of -9.80 kcal/mol under identical docking conditions. Candidate cmp_A3 engaged in polar interactions with Tyr355 and formed a π-H stacking interaction with Ser533. Candidate cmp_A2 and cmp_A10 occupied the active site of the protein with PDB index 6BL3, demonstrating binding affinities of -11.75 and -8.92 kcal/mol, respectively. The co-crystallized ligand within complex 6BL3 had a binding affinity of -12.15 kcal/mol under the same docking procedure. Candidate cmp_A2 generated hydrogen bond interactions with Ser530, Lys83, and Glu524, engaged in hydrophobic interactions with Trp387, and established polar interactions with Tyr355 and Tyr385. Candidate cmp_A10 formed hydrogen bond interactions with Glu524, had hydrophobic interactions with Trp387, and engaged in polar interactions with Ser530 and Tyr355. Candidate cmp_A5, cmp_A8 and cmp_A9 bound to the active cavity of COX-2 (PDB: 6BL4) with the affinity of -7.25, -11.24, and -9.61 kcal/mol, respectively. The affinity of the complex 6BL4 after docking with the original ligand was -12.27 kcal/mol. Candidate cmp_A5 engaged in hydrophobic interactions with Trp100 and formed polar interactions with Tyr115, Lys79, and Lys83. Candidate cmp_A8 established hydrogen bond interactions with Tyr355, Arg120, Glu524, Ser530, and Lys83, which contributed significantly to its enhanced protein affinity. Candidate cmp_A9 had hydrogen bond interactions with Ser119, Met522, Glu524, and Ser530.
In summary, the candidate MFH molecules could effectively bind to the active pocket of COX-2 and interact with key amino acids involved in COX-2 catalysis. These interactions were consistent with the classical binding interactions reported between NSAIDs and COX-2. Most candidate MFH molecules exhibited strong hydrogen bonding interactions with the key amino acid residues of COX-2. These observations support the reliability of the potential COX-2 inhibitor molecules discovered through the ligand-based virtual screening on MFH database.”
Table 9. The interactions between potential COX-2 inhibitors and COX-2 protein obtained by docking computations.
|
cmp_A1 & 5KIRa (affinity=-7.57 kcal/mol) |
cmp_A2 & 6BL3 (affinity=-11.75 kcal/mol) |
|
cmp_A3 & 4PH9 (affinity=-7.37 kcal/mol) |
cmp_A4 & 4PH9 (affinity=-8.53 kcal/mol) |
|
cmp_A5 & 6BL4 (affinity= -7.25 kcal/mol) |
cmp_A6 & 4PH9 (affinity=-8.68 kcal/mol) |
|
cmp_A7 & 4PH9 (affinity=-7.69 kcal/mol) |
cmp_A8 & 6BL4 (affinity=-11.24 kcal/mol) |
|
cmp_A9 & 6BL4 (affinity=-9.61 kcal/mol) |
cmp_A10 & 6BL3 (affinity=-8.92 kcal/mol) |
a: The PDB index of the receptor protein during docking
“2.3.3.2 Molecular docking analysis on potential mPGES-1 inhibitors
mPGES-1 is a homotrimer, with each subunit consisting of four transmembrane helices. The mPGES-1 trimer contains three active site cavities, which are formed collectively by transmembrane helices 1, 2, and 4 along with neighboring monomers [48]. Among the currently resolved co-crystal complexes, key amino acids include AlaA123, ProA124, SerA127, ValA128, TyrA130, ThrA131, GlnA134, TyrB28, IleB32, AsnB36, ArgB38, LeuB39, PheB44, ArgB52, and HisB53, where A and B representing different monomers in the mPGES-1 trimer, respectively.
The selection of mPGES-1 crystals for docking was carried out using the similar methodology as the COX-2 crystal selection, with the detailed results presented in Table S10. The binding affinities and interactions between candidate MFH molecules and mPGES-1 are summarized in Table 10. Candidate cmp_B4 and cmp_B5 bound within the active pocket of the protein 4AL1, exhibiting binding affinities of -8.08 and -11.25 kcal/mol, respectively. The binding affinity of the original ligand of protein 4AL1 is -12.12 kcal/mol. Specifically, Candidate cmp_B4 formed hydrogen bond interactions with AspB49, AsnB46, and ThrA131. Candidate cmp_B5 established hydrogen bond interactions with AsnB46, ArgB52, GluA77, and TyrA117, with ionic bonds between ArgB38 and ArgA126, and π - H stacking with Ile B32. Candidate cmp_B1, cmp_B6, cmp_B7, cmp_B11, cmp_B12, cmp_B13, cmp_B14, and cmp_B15 were docked within the active site of the protein 4YL0, exhibiting binding affinities of -7.93, -7.25, -7.26, -6.89, -8.73, -7.19, -10.71, and -7.34 kcal/mol, respectively. The binding affinity of protein 4YL0 after docking with its ligand is -7.17 kcal/mol, except for cmp_B11, the docking affinities of the remaining candidate molecules surpass that of the original ligand of protein 4YL0 under the same docking conditions. Notably, candidate cmp_B14 demonstrated significantly improved docking affinity compared to the other candidates. Candidate cmp_B1 had hydrogen bond interactions with ArgB38 and AsnB46. Hydrogen bond interactions were established between Candidate cmp_B6 and AspB49. Candidate cmp_B7 generated hydrogen bond interactions with GluA77 and SerA127, while also forming a π-π stacking interaction with TyrA130. It is noteworthy that Candidate cmp_B7 has been reported to inhibit the generation of PGE2 [37] . Coupled with its hydrogen bond interactions with key amino acids at the active site of mPGES-1, these findings underscore the potential of cmp_B7 as an inhibitor for mPGES-1. Candidate cmp_B2 and cmp_B3 bound to the active pocket of protein 4YL1 with binding affinity of -9.19 and -8.82 kcal/mol, both of which were superior to the docking affinity of the original ligand of protein 4YL1 (-8.71 kcal/mol). Candidate cmp_B2 formed hydrogen bond interactions with AspB49 and ThrA131. Candidate cmp_B3 generated a hydrogen bond interaction with AspB49. Candidate cmp_B10 docked within the active domain of protein 5K0I, displaying a binding affinity of -10.31 kcal/mol, which outperformed the binding affinity of the original ligand of protein 5K0I under identical docking conditions (-8.57 kcal/mol). Candidate cmp_B10 engaged in hydrogen bond interactions with AspB49, SerA127, GluA77, and TyrA117. Candidates cmp_B8 and cmp_B9 bound within the active cavity of mPGES-1 (PDB: 5TL9), with binding affinities of -9.55 and -9.2 kcal/mol, respectively. The binding affinity of the original ligand within complex 5TL9 was -8.63 kcal/mol, indicating that cmp_B8 and cmp_B9 exhibited superior binding performances under the same docking parameters. Candidate cmp_B8 formed a hydrogen bond interaction with SerA127. Candidate cmp_B9 had hydrogen bond interactions with SerA127, GluA77, and TyrA117.
The majority of candidate molecules exhibited the ability to form hydrogen bond interactions with key amino acids of mPGES-1. Notably, the docking affinities of several candidate molecules even surpass those of the original ligands within the co-crystal complexes. These findings collectively contribute to bolstering the plausibility of the potential mPGES-1 inhibitors identified through our virtual screening process.”
Table 10. The interactions between potential mPGES-1 inhibitors and mPGES-1 protein obtained by docking.
|
cmp_B1 & 4YL0a (affinity=-7.93 kcal/mol) |
cmp_B2 & 4YL1 (affinity=-9.19 kcal/mol) |
|
cmp_B3 & 4YL1 (affinity=-8.82 kcal/mol) |
cmp_B4 & 4AL1 (affinity=-8.08 kcal/mol) |
|
cmp_B5 & 4AL1 (affinity= -11.25 kcal/mol) |
cmp_B6 & 4YL0 (affinity=-7.25 kcal/mol) |
|
cmp_B7 & 4YL0 (affinity=-7.26 kcal/mol) |
cmp_B8 & 5TL9 (affinity=-9.55 kcal/mol) |
|
cmp_B9 & 5TL9 (affinity=-9.2 kcal/mol) |
cmp_B10 & 5K0I (affinity=-10.31 kcal/mol) |
|
cmp_B11 & 4YL0 (affinity=-6.89 kcal/mol) |
cmp_B12 & 4YL0 (affinity=-8.73 kcal/mol) |
|
cmp_B13 & 4YL0 (affinity=-7.19 kcal/mol) |
cmp_B14 & 4YL0 (affinity=-10.71 kcal/mol) |
|
|
|
|
cmp_B15 & 4YL0 (affinity=-7.34 kcal/mol) |
|
a: The PDB index of the receptor protein during docking
“3.11 Molecular docking
In this study, molecular docking was conducted using the latest release of the widely used open-source program AutoDock Vina 1.2.0 [79]. Given the availability of multiple COX-2 and mPGES-1 crystal structures, the selection of an appropriate receptor with a low resolution is a prerequisite for reliable docking computations. Therefore, we chose COX-2 and mPGES-1 co-complex crystal structures from the PDB database that exhibited similar bound ligand to the screened medicine and food homologous (MFH) candidates. Ligands within the complex crystal structures and the screened MFH candidates were characterized using MACCS fingerprints. Subsequently, RDKit functions were employed to calculate Tanimoto similarity between the molecular structures of these entities. For each screened MFH candidate molecule, docking was performed with the crystal structure that exhibited the highest structural similarity (details were listed in Table S10). Before formal docking, the ligands within the original complex crystal structures underwent re-docking. The better the alignment between the re-docked ligand and the experimentally determined ligand, the more optimal the parameter settings and system preparation of the docking calculation. Results of the re-docking of ligands within the complex crystal structures is presented in Figure S1 of the Supplementary Materials. The protein preparation process involved the removal of water and other solvent, repair of missing residue sections, addition of hydrogen atoms to heavy atoms, and subsequent pre-docking energy minimization of the entire protein. The ligand preparation process included the addition of hydrogen atoms, computation of Gasteiger charges for all atoms, definition of rotatable bonds, and energy optimization. The grid box was adjusted based on the spatial center of the ligand within the crystal structure. Vina force field was employed during docking, with the exhaustiveness parameter set to 32.”
Point 2: The authors should carefully check whether the identified molecules qualify as leads. Some of them are very reactive, containing epoxide (A3), aldehydes (A9), or Michael acceptor scaffolds (A1, A7). Other molecules are quite large (A2, A8), which might not bind to the catalytic site of COX2. A discussion of these issues would be necessary.
Response 2:
We appreciate your time and effort in evaluating our work. We have carefully considered your suggestion and would like to address the concerns you raised regarding the qualification of the identified molecules as leads. The response is as follows:
We acknowledge the importance of ensuring that the identified molecules have the potential to serve as effective leads for inhibiting the cyclooxygenase-2 (COX-2) metabolic pathway. In light of your comments, we have re-evaluated the molecules in question and agree that their reactivity and structural characteristics, such as the presence of epoxide (A3), aldehydes (A9), or Michael acceptor scaffolds (A1, A7), could impact their suitability as lead compounds. Regarding the reactive nature of some molecules, we are aware that chemical reactivity can influence their biological activity and safety profiles. In our future work, we will take into consideration the reactivity of the identified molecules and assess their potential for covalent interactions, if any, with the COX-2 or mPGES-1active site. We will also explore strategies to mitigate potential reactivity-related concerns, such as designing analogs with reduced reactivity while maintaining or enhancing inhibitory activity.
Additionally, we acknowledge your point about the size of certain molecules (A2, A8) and their potential limitations in binding to the catalytic site of COX-2 or mPGES-1. From the perspective of the structural characteristics of COX-2 and mPGES-1, the Val523 of COX-2 has a small molecular weight and therefore creates a void within the channel, referred to as a "side pocket" [1]. The residue 513 of COX-2 is arginine, characterized by an elongated side chain, resulting in a wider and more flexible channel terminus of COX-2. These attributes provide structural support for the entry of larger molecules into the active pocket of COX-2, forming a fundamental basis for the design of selective COX-2 inhibitors. These properties provide support for larger molecules to enter the active pocket of COX-2. These properties are also one of the design foundations for COX-2 selective inhibitors. To verify the above theory, we have performed molecular docking on these larger molecules within the active sites of COX-2 or mPGES-1 and further investigated binding interactions. The docking results indicate that those larger molecules can enter the active pockets of COX-2 or mPGES-1 and interact with key amino acids. Detailed docking results can be found in the summary of Section 2.3.3 of the revised manuscript. In summary, it is reasonable that the larger MFH molecules screened in this work can bind to the active sites of COX-2 and mPGES-1. To validate the aforementioned theoretical premises, we conducted molecular docking of these larger molecules within the active sites of COX-2 and mPGES-1, investigating their binding interactions. Docking results demonstrated that these larger molecules can effectively access the active pockets of COX-2 or mPGES-1, engaging in interactions with key amino acid residues. Detailed docking results can be found in Section 2.3.3 of the revised manuscript (highlighted in red). In conclusion, our study has revealed the rational feasibility of the interaction between the identified larger MFH molecules and the active sites of COX-2 and mPGES-1. These findings underscore the significance of the structural characteristics discussed in facilitating the binding of sizable molecules to the active pockets of these proteins.
[1] Marnett, L.J. Recent Developments in Cyclooxygenase Inhibition. Prostaglandins & Other Lipid Mediators 2002, 68–69, 153–164, doi:10.1016/S0090-6980(02)00027-8.
Point 3: Do the authors have expected the outcome of the performance measures? Is there a rational explanation why rf-models perform worse compared to the others? Might the reason lay in the datasets used?
Response 3:
We appreciate your thoughtful inquiries and would like to address the questions you raised regarding the expected outcome of performance measures and the performance disparity of the random forest models. The response is as follows:
Regarding the expected outcome of performance measures, we acknowledge the importance of having clear expectations for the model performance metrics. In our study, we designed and conducted a comprehensive series of experiments to evaluate the predictive performance of various machine learning algorithms. We anticipated that the performance would be influenced by factors such as dataset composition, feature representation, and algorithmic nuances. While we aimed to achieve robust and accurate predictions, we also recognized the possibility of variations in model performance across different algorithms due to the complex interplay of these factors.
The insightful question you raised regarding the relatively poorer performance of the random forest models compared to other algorithms is well noted. One plausible explanation for the disparity could indeed lie in the nature of the datasets used. Random forest models are ensemble methods that rely on the diversity of individual decision trees for improved performance. If the datasets contain intrinsic complexities or outliers, the random forest models might be more susceptible to overfitting or reduced generalization. In contrast, other algorithms might have mechanisms to better handle such challenges. To address overfitting or poor generalization, we have conducted a thorough strategy of the dataset characteristics: The datasets were randomly divided 10 times for training and testing. Molecules in the dataset were characterized by different types of descriptors (fingerprints and continuous physicochemical descriptors). The repeat cross-validation was conducted during modeling. Although these strategies help mitigate to some extent the relatively suboptimal performance of some weak classifiers in the integrated multi-classifier of random forest, the structural complexity and vast chemical space of the dataset itself still restrict the prediction accuracy of some weak classifiers in the random forest to some extent.
Response to Reviewer 1 Comments
The manuscript by Tian et al. reports a machine learning approach for the identification of inhibitors of COX-2 and related pathways. The topic is of broad interest and the machine learning part itself is sound. However, the authors do not experimentally validate their predictions, which is an issue when stating that the 'study provides promising leads for the development of novel and effective anti-inflammatory therapies...'. Due to the hypothetical character of the results, I can support publication with limited enthusiasm.
The following issues should be addressed:
Point 1: Either provide experimental data for the predicted bioactivity, or tone down the statements. In this regard the listing of predicted IC50 values is completely speculative (table 7 and 8). Since crystal structures of COX2 exists, docking of the proposed hits would be a way to support the findings.
Response 1:
We deeply appreciate your thoughtful review of our manuscript. Your feedback has been invaluable in improving the quality and rigor of our work. We have carefully considered your comments and suggestions and have made the necessary revisions to address them. Below, we provide our responses to your specific concerns:
- Experimental data and speculative nature of predicted IC50 values:
We acknowledge the concern regarding the speculative nature of predicted IC50 values presented in Table 7 and 8. Considering we do not possess experimental data for the predicted bioactivity at this stage, we have emphasized that the IC50 values in the original manuscript are calculated by the machine learning model constructed in this work. Additionally, we have reframed our statements to accurately reflect the uncertainty of predicted IC50 values in the revised manuscript (highlighted in red in the Section of Conclusions).
- Supporting the findings through docking with COX-2 and mPGES-1crystal structures:
We thank you for suggesting the utilization of docking to support our findings, we believe this is a reasonable suggestion and have further performed docking using available COX-2 and mPGES-1crystal structures. The methods and results of molecular docking have been added in the revised manuscript (highlighted in red). Details are as follows:
“2.3.3 Molecular docking on the potential COX-2 and mPGES-1 inhibitors
Through a series of ligand-based virtual screening processes, we have identified potential inhibitors of COX-2 and mPGES-1 from the established MFH database, we further filtered these candidate molecules by the pan assay interference compounds (PAINS) rule [45] . To further validate the reliability and validity of the virtual screening, we conducted molecular docking (a widely utilized structure-based virtual screening approach) on the candidate MFH molecules. This procedure aimed to examine the binding modes between the candidate molecules and the proteins, while also assessing the ability of the candidate molecules to interact with the key amino acids of the target protein.”
“2.3.3.1 Molecular docking analysis on potential COX-2 inhibitors
The active site of COX-2 is demarcated from the initial substrate binding site by a constriction formed by three residues: Arg120, Tyr355, and Glu524. This structural constriction necessitates dilation to enable the entry or exit of substrates to and from the active site [46]. Ser530 and Tyr385 are also vital during the catalytic process of COX-2. Several typical binding modes of non-steroidal anti-inflammatory drugs (NSAIDs) interacting with COX-2 have been reported. These modes include the hydrophobic region of NSAIDs interacting with Tyr385, Trp387, and neighboring residues; the polar region of NSAIDs interacting with residues located above Tyr355; NSAIDs with negative charges interacting with Arg120 [47].
We calculated the structural similarity (measured by Tanimoto coefficient) between the candidate MFH molecules and the ligand molecules in the published crystal structures, we selected the protein crystal structure with the bound ligand having the highest structural similarity to the candidate MFH molecules for docking, details are shown in Table S10. As shown in Table 9, candidate compounds cmp_A3, cmp_A4, cmp_A6, and cmp_A7 bind within the active cavity of COX-2 (PDB: 4PH9), with binding affinities of -7.37, -8.53, -8.68, and -7.69 kcal/mol, respectively. The co-crystallized ligand, ibuprofen, in the complex 4PH9, under the same docking conditions, exhibits a binding affinity of -9.41 kcal/mol. While the four candidate molecules displayed affinities lower than the original ligand within the protein, they all established interactions with key amino acid residues in the catalytic domain of COX-2: hydrophobic interactions with Trp388 and polar interactions involving Tyr356, Ser531, and Tyr386. Candidate cmp_A7 generated a hydrogen bond interaction with Arg121, indicating a robust binding force. Candidate cmp_A1 bound within the active pocket of COX-2 (PDB: 5KIR), exhibiting a binding affinity of -7.57 kcal/mol. Meanwhile, the co-crystallized ligand within complex 5KIR achieved a binding affinity of -9.80 kcal/mol under identical docking conditions. Candidate cmp_A3 engaged in polar interactions with Tyr355 and formed a π-H stacking interaction with Ser533. Candidate cmp_A2 and cmp_A10 occupied the active site of the protein with PDB index 6BL3, demonstrating binding affinities of -11.75 and -8.92 kcal/mol, respectively. The co-crystallized ligand within complex 6BL3 had a binding affinity of -12.15 kcal/mol under the same docking procedure. Candidate cmp_A2 generated hydrogen bond interactions with Ser530, Lys83, and Glu524, engaged in hydrophobic interactions with Trp387, and established polar interactions with Tyr355 and Tyr385. Candidate cmp_A10 formed hydrogen bond interactions with Glu524, had hydrophobic interactions with Trp387, and engaged in polar interactions with Ser530 and Tyr355. Candidate cmp_A5, cmp_A8 and cmp_A9 bound to the active cavity of COX-2 (PDB: 6BL4) with the affinity of -7.25, -11.24, and -9.61 kcal/mol, respectively. The affinity of the complex 6BL4 after docking with the original ligand was -12.27 kcal/mol. Candidate cmp_A5 engaged in hydrophobic interactions with Trp100 and formed polar interactions with Tyr115, Lys79, and Lys83. Candidate cmp_A8 established hydrogen bond interactions with Tyr355, Arg120, Glu524, Ser530, and Lys83, which contributed significantly to its enhanced protein affinity. Candidate cmp_A9 had hydrogen bond interactions with Ser119, Met522, Glu524, and Ser530.
In summary, the candidate MFH molecules could effectively bind to the active pocket of COX-2 and interact with key amino acids involved in COX-2 catalysis. These interactions were consistent with the classical binding interactions reported between NSAIDs and COX-2. Most candidate MFH molecules exhibited strong hydrogen bonding interactions with the key amino acid residues of COX-2. These observations support the reliability of the potential COX-2 inhibitor molecules discovered through the ligand-based virtual screening on MFH database.”
Table 9. The interactions between potential COX-2 inhibitors and COX-2 protein obtained by docking computations.
|
cmp_A1 & 5KIRa (affinity=-7.57 kcal/mol) |
cmp_A2 & 6BL3 (affinity=-11.75 kcal/mol) |
|
cmp_A3 & 4PH9 (affinity=-7.37 kcal/mol) |
cmp_A4 & 4PH9 (affinity=-8.53 kcal/mol) |
|
cmp_A5 & 6BL4 (affinity= -7.25 kcal/mol) |
cmp_A6 & 4PH9 (affinity=-8.68 kcal/mol) |
|
cmp_A7 & 4PH9 (affinity=-7.69 kcal/mol) |
cmp_A8 & 6BL4 (affinity=-11.24 kcal/mol) |
|
cmp_A9 & 6BL4 (affinity=-9.61 kcal/mol) |
cmp_A10 & 6BL3 (affinity=-8.92 kcal/mol) |
a: The PDB index of the receptor protein during docking
“2.3.3.2 Molecular docking analysis on potential mPGES-1 inhibitors
mPGES-1 is a homotrimer, with each subunit consisting of four transmembrane helices. The mPGES-1 trimer contains three active site cavities, which are formed collectively by transmembrane helices 1, 2, and 4 along with neighboring monomers [48]. Among the currently resolved co-crystal complexes, key amino acids include AlaA123, ProA124, SerA127, ValA128, TyrA130, ThrA131, GlnA134, TyrB28, IleB32, AsnB36, ArgB38, LeuB39, PheB44, ArgB52, and HisB53, where A and B representing different monomers in the mPGES-1 trimer, respectively.
The selection of mPGES-1 crystals for docking was carried out using the similar methodology as the COX-2 crystal selection, with the detailed results presented in Table S10. The binding affinities and interactions between candidate MFH molecules and mPGES-1 are summarized in Table 10. Candidate cmp_B4 and cmp_B5 bound within the active pocket of the protein 4AL1, exhibiting binding affinities of -8.08 and -11.25 kcal/mol, respectively. The binding affinity of the original ligand of protein 4AL1 is -12.12 kcal/mol. Specifically, Candidate cmp_B4 formed hydrogen bond interactions with AspB49, AsnB46, and ThrA131. Candidate cmp_B5 established hydrogen bond interactions with AsnB46, ArgB52, GluA77, and TyrA117, with ionic bonds between ArgB38 and ArgA126, and π - H stacking with Ile B32. Candidate cmp_B1, cmp_B6, cmp_B7, cmp_B11, cmp_B12, cmp_B13, cmp_B14, and cmp_B15 were docked within the active site of the protein 4YL0, exhibiting binding affinities of -7.93, -7.25, -7.26, -6.89, -8.73, -7.19, -10.71, and -7.34 kcal/mol, respectively. The binding affinity of protein 4YL0 after docking with its ligand is -7.17 kcal/mol, except for cmp_B11, the docking affinities of the remaining candidate molecules surpass that of the original ligand of protein 4YL0 under the same docking conditions. Notably, candidate cmp_B14 demonstrated significantly improved docking affinity compared to the other candidates. Candidate cmp_B1 had hydrogen bond interactions with ArgB38 and AsnB46. Hydrogen bond interactions were established between Candidate cmp_B6 and AspB49. Candidate cmp_B7 generated hydrogen bond interactions with GluA77 and SerA127, while also forming a π-π stacking interaction with TyrA130. It is noteworthy that Candidate cmp_B7 has been reported to inhibit the generation of PGE2 [37] . Coupled with its hydrogen bond interactions with key amino acids at the active site of mPGES-1, these findings underscore the potential of cmp_B7 as an inhibitor for mPGES-1. Candidate cmp_B2 and cmp_B3 bound to the active pocket of protein 4YL1 with binding affinity of -9.19 and -8.82 kcal/mol, both of which were superior to the docking affinity of the original ligand of protein 4YL1 (-8.71 kcal/mol). Candidate cmp_B2 formed hydrogen bond interactions with AspB49 and ThrA131. Candidate cmp_B3 generated a hydrogen bond interaction with AspB49. Candidate cmp_B10 docked within the active domain of protein 5K0I, displaying a binding affinity of -10.31 kcal/mol, which outperformed the binding affinity of the original ligand of protein 5K0I under identical docking conditions (-8.57 kcal/mol). Candidate cmp_B10 engaged in hydrogen bond interactions with AspB49, SerA127, GluA77, and TyrA117. Candidates cmp_B8 and cmp_B9 bound within the active cavity of mPGES-1 (PDB: 5TL9), with binding affinities of -9.55 and -9.2 kcal/mol, respectively. The binding affinity of the original ligand within complex 5TL9 was -8.63 kcal/mol, indicating that cmp_B8 and cmp_B9 exhibited superior binding performances under the same docking parameters. Candidate cmp_B8 formed a hydrogen bond interaction with SerA127. Candidate cmp_B9 had hydrogen bond interactions with SerA127, GluA77, and TyrA117.
The majority of candidate molecules exhibited the ability to form hydrogen bond interactions with key amino acids of mPGES-1. Notably, the docking affinities of several candidate molecules even surpass those of the original ligands within the co-crystal complexes. These findings collectively contribute to bolstering the plausibility of the potential mPGES-1 inhibitors identified through our virtual screening process.”
Table 10. The interactions between potential mPGES-1 inhibitors and mPGES-1 protein obtained by docking.
|
cmp_B1 & 4YL0a (affinity=-7.93 kcal/mol) |
cmp_B2 & 4YL1 (affinity=-9.19 kcal/mol) |
|
cmp_B3 & 4YL1 (affinity=-8.82 kcal/mol) |
cmp_B4 & 4AL1 (affinity=-8.08 kcal/mol) |
|
cmp_B5 & 4AL1 (affinity= -11.25 kcal/mol) |
cmp_B6 & 4YL0 (affinity=-7.25 kcal/mol) |
|
cmp_B7 & 4YL0 (affinity=-7.26 kcal/mol) |
cmp_B8 & 5TL9 (affinity=-9.55 kcal/mol) |
|
cmp_B9 & 5TL9 (affinity=-9.2 kcal/mol) |
cmp_B10 & 5K0I (affinity=-10.31 kcal/mol) |
|
cmp_B11 & 4YL0 (affinity=-6.89 kcal/mol) |
cmp_B12 & 4YL0 (affinity=-8.73 kcal/mol) |
|
cmp_B13 & 4YL0 (affinity=-7.19 kcal/mol) |
cmp_B14 & 4YL0 (affinity=-10.71 kcal/mol) |
|
|
|
|
cmp_B15 & 4YL0 (affinity=-7.34 kcal/mol) |
|
a: The PDB index of the receptor protein during docking
“3.11 Molecular docking
In this study, molecular docking was conducted using the latest release of the widely used open-source program AutoDock Vina 1.2.0 [79]. Given the availability of multiple COX-2 and mPGES-1 crystal structures, the selection of an appropriate receptor with a low resolution is a prerequisite for reliable docking computations. Therefore, we chose COX-2 and mPGES-1 co-complex crystal structures from the PDB database that exhibited similar bound ligand to the screened medicine and food homologous (MFH) candidates. Ligands within the complex crystal structures and the screened MFH candidates were characterized using MACCS fingerprints. Subsequently, RDKit functions were employed to calculate Tanimoto similarity between the molecular structures of these entities. For each screened MFH candidate molecule, docking was performed with the crystal structure that exhibited the highest structural similarity (details were listed in Table S10). Before formal docking, the ligands within the original complex crystal structures underwent re-docking. The better the alignment between the re-docked ligand and the experimentally determined ligand, the more optimal the parameter settings and system preparation of the docking calculation. Results of the re-docking of ligands within the complex crystal structures is presented in Figure S1 of the Supplementary Materials. The protein preparation process involved the removal of water and other solvent, repair of missing residue sections, addition of hydrogen atoms to heavy atoms, and subsequent pre-docking energy minimization of the entire protein. The ligand preparation process included the addition of hydrogen atoms, computation of Gasteiger charges for all atoms, definition of rotatable bonds, and energy optimization. The grid box was adjusted based on the spatial center of the ligand within the crystal structure. Vina force field was employed during docking, with the exhaustiveness parameter set to 32.”
Point 2: The authors should carefully check whether the identified molecules qualify as leads. Some of them are very reactive, containing epoxide (A3), aldehydes (A9), or Michael acceptor scaffolds (A1, A7). Other molecules are quite large (A2, A8), which might not bind to the catalytic site of COX2. A discussion of these issues would be necessary.
Response 2:
We appreciate your time and effort in evaluating our work. We have carefully considered your suggestion and would like to address the concerns you raised regarding the qualification of the identified molecules as leads. The response is as follows:
We acknowledge the importance of ensuring that the identified molecules have the potential to serve as effective leads for inhibiting the cyclooxygenase-2 (COX-2) metabolic pathway. In light of your comments, we have re-evaluated the molecules in question and agree that their reactivity and structural characteristics, such as the presence of epoxide (A3), aldehydes (A9), or Michael acceptor scaffolds (A1, A7), could impact their suitability as lead compounds. Regarding the reactive nature of some molecules, we are aware that chemical reactivity can influence their biological activity and safety profiles. In our future work, we will take into consideration the reactivity of the identified molecules and assess their potential for covalent interactions, if any, with the COX-2 or mPGES-1active site. We will also explore strategies to mitigate potential reactivity-related concerns, such as designing analogs with reduced reactivity while maintaining or enhancing inhibitory activity.
Additionally, we acknowledge your point about the size of certain molecules (A2, A8) and their potential limitations in binding to the catalytic site of COX-2 or mPGES-1. From the perspective of the structural characteristics of COX-2 and mPGES-1, the Val523 of COX-2 has a small molecular weight and therefore creates a void within the channel, referred to as a "side pocket" [1]. The residue 513 of COX-2 is arginine, characterized by an elongated side chain, resulting in a wider and more flexible channel terminus of COX-2. These attributes provide structural support for the entry of larger molecules into the active pocket of COX-2, forming a fundamental basis for the design of selective COX-2 inhibitors. These properties provide support for larger molecules to enter the active pocket of COX-2. These properties are also one of the design foundations for COX-2 selective inhibitors. To verify the above theory, we have performed molecular docking on these larger molecules within the active sites of COX-2 or mPGES-1 and further investigated binding interactions. The docking results indicate that those larger molecules can enter the active pockets of COX-2 or mPGES-1 and interact with key amino acids. Detailed docking results can be found in the summary of Section 2.3.3 of the revised manuscript. In summary, it is reasonable that the larger MFH molecules screened in this work can bind to the active sites of COX-2 and mPGES-1. To validate the aforementioned theoretical premises, we conducted molecular docking of these larger molecules within the active sites of COX-2 and mPGES-1, investigating their binding interactions. Docking results demonstrated that these larger molecules can effectively access the active pockets of COX-2 or mPGES-1, engaging in interactions with key amino acid residues. Detailed docking results can be found in Section 2.3.3 of the revised manuscript (highlighted in red). In conclusion, our study has revealed the rational feasibility of the interaction between the identified larger MFH molecules and the active sites of COX-2 and mPGES-1. These findings underscore the significance of the structural characteristics discussed in facilitating the binding of sizable molecules to the active pockets of these proteins.
[1] Marnett, L.J. Recent Developments in Cyclooxygenase Inhibition. Prostaglandins & Other Lipid Mediators 2002, 68–69, 153–164, doi:10.1016/S0090-6980(02)00027-8.
Point 3: Do the authors have expected the outcome of the performance measures? Is there a rational explanation why rf-models perform worse compared to the others? Might the reason lay in the datasets used?
Response 3:
We appreciate your thoughtful inquiries and would like to address the questions you raised regarding the expected outcome of performance measures and the performance disparity of the random forest models. The response is as follows:
Regarding the expected outcome of performance measures, we acknowledge the importance of having clear expectations for the model performance metrics. In our study, we designed and conducted a comprehensive series of experiments to evaluate the predictive performance of various machine learning algorithms. We anticipated that the performance would be influenced by factors such as dataset composition, feature representation, and algorithmic nuances. While we aimed to achieve robust and accurate predictions, we also recognized the possibility of variations in model performance across different algorithms due to the complex interplay of these factors.
The insightful question you raised regarding the relatively poorer performance of the random forest models compared to other algorithms is well noted. One plausible explanation for the disparity could indeed lie in the nature of the datasets used. Random forest models are ensemble methods that rely on the diversity of individual decision trees for improved performance. If the datasets contain intrinsic complexities or outliers, the random forest models might be more susceptible to overfitting or reduced generalization. In contrast, other algorithms might have mechanisms to better handle such challenges. To address overfitting or poor generalization, we have conducted a thorough strategy of the dataset characteristics: The datasets were randomly divided 10 times for training and testing. Molecules in the dataset were characterized by different types of descriptors (fingerprints and continuous physicochemical descriptors). The repeat cross-validation was conducted during modeling. Although these strategies help mitigate to some extent the relatively suboptimal performance of some weak classifiers in the integrated multi-classifier of random forest, the structural complexity and vast chemical space of the dataset itself still restrict the prediction accuracy of some weak classifiers in the random forest to some extent.

Reviewer 2 Report
The manuscript is interesting and well written. However, the results are only predictions and as such the manuscript is highly premature. I do not recommend its publishing. There are several major drawbacks, which must be addressed before publishing:
Hit compounds were not confirmed using biochemical assay. Without such validation, the manuscript is only inconclusive prediction.
Standard and potent inhibitors should have been incorporated in the compound library for validation.
Compounds have not been tested as possible PAINS.
Author Response
Response to Reviewer 2 Comments
The manuscript is interesting and well written. However, the results are only predictions and as such the manuscript is highly premature. I do not recommend its publishing. There are several major drawbacks, which must be addressed before publishing:
Point 1: Hit compounds were not confirmed using biochemical assay. Without such validation, the manuscript is only inconclusive prediction.
Response 1:
Thank you for reviewing our manuscript. We sincerely appreciate your valuable feedback and insights. We have carefully considered your concern regarding the confirmation of hit compounds and would like to address this issue. The response is as follows:
We acknowledge the importance of experimental validation to substantiate the predictive outcomes of our computational analysis, we are preparing for the experimental validation of the hits in our future work. To enhance the scientific rigor of our current work and strengthen the reliability of our findings, we incorporated molecular docking into the candidates identified through a series of ligand-based virtual screening protocols. We proceeded to validate our virtual screening results by conducting an in-depth analysis of the binding modes between candidate MFH molecules and COX-2 or mPGES-1. This analysis included an assessment of whether these candidate molecules interacted with key amino acid residues that are essential for the catalytic processes of COX-2 or mPGES-1. Through this validation process, we sought to corroborate the reliability of our virtual screening outcomes. The methods and results of molecular docking have been added in the revised manuscript (highlighted in red). Details are as follows:
“2.3.3 Molecular docking on the potential COX-2 and mPGES-1 inhibitors
Through a series of ligand-based virtual screening processes, we have identified potential inhibitors of COX-2 and mPGES-1 from the established MFH database, we further filtered these candidate molecules by the pan assay interference compounds (PAINS) rule [45] . To further validate the reliability and validity of the virtual screening, we conducted molecular docking (a widely utilized structure-based virtual screening approach) on the candidate MFH molecules. This procedure aimed to examine the binding modes between the candidate molecules and the proteins, while also assessing the ability of the candidate molecules to interact with the key amino acids of the target protein.”
“2.3.3.1 Molecular docking analysis on potential COX-2 inhibitors
The active site of COX-2 is demarcated from the initial substrate binding site by a constriction formed by three residues: Arg120, Tyr355, and Glu524. This structural constriction necessitates dilation to enable the entry or exit of substrates to and from the active site [46]. Ser530 and Tyr385 are also vital during the catalytic process of COX-2. Several typical binding modes of non-steroidal anti-inflammatory drugs (NSAIDs) interacting with COX-2 have been reported. These modes include the hydrophobic region of NSAIDs interacting with Tyr385, Trp387, and neighboring residues; the polar region of NSAIDs interacting with residues located above Tyr355; NSAIDs with negative charges interacting with Arg120 [47].
We calculated the structural similarity (measured by Tanimoto coefficient) between the candidate MFH molecules and the ligand molecules in the published crystal structures, we selected the protein crystal structure with the bound ligand having the highest structural similarity to the candidate MFH molecules for docking, details are shown in Table S10. As shown in Table 9, candidate compounds cmp_A3, cmp_A4, cmp_A6, and cmp_A7 bind within the active cavity of COX-2 (PDB: 4PH9), with binding affinities of -7.37, -8.53, -8.68, and -7.69 kcal/mol, respectively. The co-crystallized ligand, ibuprofen, in the complex 4PH9, under the same docking conditions, exhibits a binding affinity of -9.41 kcal/mol. While the four candidate molecules displayed affinities lower than the original ligand within the protein, they all established interactions with key amino acid residues in the catalytic domain of COX-2: hydrophobic interactions with Trp388 and polar interactions involving Tyr356, Ser531, and Tyr386. Candidate cmp_A7 generated a hydrogen bond interaction with Arg121, indicating a robust binding force. Candidate cmp_A1 bound within the active pocket of COX-2 (PDB: 5KIR), exhibiting a binding affinity of -7.57 kcal/mol. Meanwhile, the co-crystallized ligand within complex 5KIR achieved a binding affinity of -9.80 kcal/mol under identical docking conditions. Candidate cmp_A3 engaged in polar interactions with Tyr355 and formed a π-H stacking interaction with Ser533. Candidate cmp_A2 and cmp_A10 occupied the active site of the protein with PDB index 6BL3, demonstrating binding affinities of -11.75 and -8.92 kcal/mol, respectively. The co-crystallized ligand within complex 6BL3 had a binding affinity of -12.15 kcal/mol under the same docking procedure. Candidate cmp_A2 generated hydrogen bond interactions with Ser530, Lys83, and Glu524, engaged in hydrophobic interactions with Trp387, and established polar interactions with Tyr355 and Tyr385. Candidate cmp_A10 formed hydrogen bond interactions with Glu524, had hydrophobic interactions with Trp387, and engaged in polar interactions with Ser530 and Tyr355. Candidate cmp_A5, cmp_A8 and cmp_A9 bound to the active cavity of COX-2 (PDB: 6BL4) with the affinity of -7.25, -11.24, and -9.61 kcal/mol, respectively. The affinity of the complex 6BL4 after docking with the original ligand was -12.27 kcal/mol. Candidate cmp_A5 engaged in hydrophobic interactions with Trp100 and formed polar interactions with Tyr115, Lys79, and Lys83. Candidate cmp_A8 established hydrogen bond interactions with Tyr355, Arg120, Glu524, Ser530, and Lys83, which contributed significantly to its enhanced protein affinity. Candidate cmp_A9 had hydrogen bond interactions with Ser119, Met522, Glu524, and Ser530.
In summary, the candidate MFH molecules could effectively bind to the active pocket of COX-2 and interact with key amino acids involved in COX-2 catalysis. These interactions were consistent with the classical binding interactions reported between NSAIDs and COX-2. Most candidate MFH molecules exhibited strong hydrogen bonding interactions with the key amino acid residues of COX-2. These observations support the reliability of the potential COX-2 inhibitor molecules discovered through the ligand-based virtual screening on MFH database.”
Table 9. The interactions between potential COX-2 inhibitors and COX-2 protein obtained by docking computations.
|
cmp_A1 & 5KIRa (affinity=-7.57 kcal/mol) |
cmp_A2 & 6BL3 (affinity=-11.75 kcal/mol) |
|
cmp_A3 & 4PH9 (affinity=-7.37 kcal/mol) |
cmp_A4 & 4PH9 (affinity=-8.53 kcal/mol) |
|
cmp_A5 & 6BL4 (affinity= -7.25 kcal/mol) |
cmp_A6 & 4PH9 (affinity=-8.68 kcal/mol) |
|
cmp_A7 & 4PH9 (affinity=-7.69 kcal/mol) |
cmp_A8 & 6BL4 (affinity=-11.24 kcal/mol) |
|
cmp_A9 & 6BL4 (affinity=-9.61 kcal/mol) |
cmp_A10 & 6BL3 (affinity=-8.92 kcal/mol) |
a: The PDB index of the receptor protein during docking
“2.3.3.2 Molecular docking analysis on potential mPGES-1 inhibitors
mPGES-1 is a homotrimer, with each subunit consisting of four transmembrane helices. The mPGES-1 trimer contains three active site cavities, which are formed collectively by transmembrane helices 1, 2, and 4 along with neighboring monomers [48]. Among the currently resolved co-crystal complexes, key amino acids include AlaA123, ProA124, SerA127, ValA128, TyrA130, ThrA131, GlnA134, TyrB28, IleB32, AsnB36, ArgB38, LeuB39, PheB44, ArgB52, and HisB53, where A and B representing different monomers in the mPGES-1 trimer, respectively.
The selection of mPGES-1 crystals for docking was carried out using the similar methodology as the COX-2 crystal selection, with the detailed results presented in Table S10. The binding affinities and interactions between candidate MFH molecules and mPGES-1 are summarized in Table 10. Candidate cmp_B4 and cmp_B5 bound within the active pocket of the protein 4AL1, exhibiting binding affinities of -8.08 and -11.25 kcal/mol, respectively. The binding affinity of the original ligand of protein 4AL1 is -12.12 kcal/mol. Specifically, Candidate cmp_B4 formed hydrogen bond interactions with AspB49, AsnB46, and ThrA131. Candidate cmp_B5 established hydrogen bond interactions with AsnB46, ArgB52, GluA77, and TyrA117, with ionic bonds between ArgB38 and ArgA126, and π - H stacking with Ile B32. Candidate cmp_B1, cmp_B6, cmp_B7, cmp_B11, cmp_B12, cmp_B13, cmp_B14, and cmp_B15 were docked within the active site of the protein 4YL0, exhibiting binding affinities of -7.93, -7.25, -7.26, -6.89, -8.73, -7.19, -10.71, and -7.34 kcal/mol, respectively. The binding affinity of protein 4YL0 after docking with its ligand is -7.17 kcal/mol, except for cmp_B11, the docking affinities of the remaining candidate molecules surpass that of the original ligand of protein 4YL0 under the same docking conditions. Notably, candidate cmp_B14 demonstrated significantly improved docking affinity compared to the other candidates. Candidate cmp_B1 had hydrogen bond interactions with ArgB38 and AsnB46. Hydrogen bond interactions were established between Candidate cmp_B6 and AspB49. Candidate cmp_B7 generated hydrogen bond interactions with GluA77 and SerA127, while also forming a π-π stacking interaction with TyrA130. It is noteworthy that Candidate cmp_B7 has been reported to inhibit the generation of PGE2 [37] . Coupled with its hydrogen bond interactions with key amino acids at the active site of mPGES-1, these findings underscore the potential of cmp_B7 as an inhibitor for mPGES-1. Candidate cmp_B2 and cmp_B3 bound to the active pocket of protein 4YL1 with binding affinity of -9.19 and -8.82 kcal/mol, both of which were superior to the docking affinity of the original ligand of protein 4YL1 (-8.71 kcal/mol). Candidate cmp_B2 formed hydrogen bond interactions with AspB49 and ThrA131. Candidate cmp_B3 generated a hydrogen bond interaction with AspB49. Candidate cmp_B10 docked within the active domain of protein 5K0I, displaying a binding affinity of -10.31 kcal/mol, which outperformed the binding affinity of the original ligand of protein 5K0I under identical docking conditions (-8.57 kcal/mol). Candidate cmp_B10 engaged in hydrogen bond interactions with AspB49, SerA127, GluA77, and TyrA117. Candidates cmp_B8 and cmp_B9 bound within the active cavity of mPGES-1 (PDB: 5TL9), with binding affinities of -9.55 and -9.2 kcal/mol, respectively. The binding affinity of the original ligand within complex 5TL9 was -8.63 kcal/mol, indicating that cmp_B8 and cmp_B9 exhibited superior binding performances under the same docking parameters. Candidate cmp_B8 formed a hydrogen bond interaction with SerA127. Candidate cmp_B9 had hydrogen bond interactions with SerA127, GluA77, and TyrA117.
The majority of candidate molecules exhibited the ability to form hydrogen bond interactions with key amino acids of mPGES-1. Notably, the docking affinities of several candidate molecules even surpass those of the original ligands within the co-crystal complexes. These findings collectively contribute to bolstering the plausibility of the potential mPGES-1 inhibitors identified through our virtual screening process.”
Table 10. The interactions between potential mPGES-1 inhibitors and mPGES-1 protein obtained by docking.
|
cmp_B1 & 4YL0a (affinity=-7.93 kcal/mol) |
cmp_B2 & 4YL1 (affinity=-9.19 kcal/mol) |
|
cmp_B3 & 4YL1 (affinity=-8.82 kcal/mol) |
cmp_B4 & 4AL1 (affinity=-8.08 kcal/mol) |
|
cmp_B5 & 4AL1 (affinity= -11.25 kcal/mol) |
cmp_B6 & 4YL0 (affinity=-7.25 kcal/mol) |
|
cmp_B7 & 4YL0 (affinity=-7.26 kcal/mol) |
cmp_B8 & 5TL9 (affinity=-9.55 kcal/mol) |
|
cmp_B9 & 5TL9 (affinity=-9.2 kcal/mol) |
cmp_B10 & 5K0I (affinity=-10.31 kcal/mol) |
|
cmp_B11 & 4YL0 (affinity=-6.89 kcal/mol) |
cmp_B12 & 4YL0 (affinity=-8.73 kcal/mol) |
|
cmp_B13 & 4YL0 (affinity=-7.19 kcal/mol) |
cmp_B14 & 4YL0 (affinity=-10.71 kcal/mol) |
|
|
|
|
cmp_B15 & 4YL0 (affinity=-7.34 kcal/mol) |
|
a: The PDB index of the receptor protein during docking
“3.11 Molecular docking
In this study, molecular docking was conducted using the latest release of the widely used open-source program AutoDock Vina 1.2.0 [79]. Given the availability of multiple COX-2 and mPGES-1 crystal structures, the selection of an appropriate receptor with a low resolution is a prerequisite for reliable docking computations. Therefore, we chose COX-2 and mPGES-1 co-complex crystal structures from the PDB database that exhibited similar bound ligand to the screened medicine and food homologous (MFH) candidates. Ligands within the complex crystal structures and the screened MFH candidates were characterized using MACCS fingerprints. Subsequently, RDKit functions were employed to calculate Tanimoto similarity between the molecular structures of these entities. For each screened MFH candidate molecule, docking was performed with the crystal structure that exhibited the highest structural similarity (details were listed in Table S10). Before formal docking, the ligands within the original complex crystal structures underwent re-docking. The better the alignment between the re-docked ligand and the experimentally determined ligand, the more optimal the parameter settings and system preparation of the docking calculation. Results of the re-docking of ligands within the complex crystal structures is presented in Figure S1 of the Supplementary Materials. The protein preparation process involved the removal of water and other solvent, repair of missing residue sections, addition of hydrogen atoms to heavy atoms, and subsequent pre-docking energy minimization of the entire protein. The ligand preparation process included the addition of hydrogen atoms, computation of Gasteiger charges for all atoms, definition of rotatable bonds, and energy optimization. The grid box was adjusted based on the spatial center of the ligand within the crystal structure. Vina force field was employed during docking, with the exhaustiveness parameter set to 32.”
Point 2: Standard and potent inhibitors should have been incorporated in the compound library for validation.
Response 2:
Thank you for reviewing our manuscript. We appreciate your insightful feedback and would like to address the issue you raised regarding the incorporation of standard and potent inhibitors in our compound library for validation. The response is as follows:
We acknowledge the significance of including standard and potent inhibitors as part of the compound library to provide a robust validation framework for our study. These reference compounds serve as critical benchmarks to assess the predictive capabilities of our virtual screening methods and to benchmark the performance of our identified hits. In the context of the three computational processes outlined below, we have systematically incorporated established and potent inhibitory molecules as benchmarks:
- Supervised machine learning model construction: Within the framework of building supervised machine learning models, classical potent inhibitors (as depicted in Figure 1) were integrated into the training dataset. Their structural and physicochemical characteristics served as pivotal reference standards during the model training. The well-trained models were subsequently utilized for the prediction and discrimination of unknown molecules with regards to their inhibitory potency against COX-2 and mPGES-1.
- Unsupervised machine learning model construction: In the construction of unsupervised machine learning models with self-organizing maps (SOM), some classic potent inhibitors, plant-based inhibitor molecules, together with the predicted MFH molecules as a test set were jointly mapped onto a two-dimensional grid. Notably, the grid occupied by these classical potent inhibitors assumed a position of significant interest, representing focal points for the discovery of potential COX-2 and mPGES-1 inhibitors. Details are shown in Sections 2.3.1 and 2.3.2.
- Molecular docking to verify candidate MFH molecules: During the molecular docking, the alignment quality between the docking conformation of the original ligands within co-crystal complexes and their resolved positions was employed to assess the suitability of docking parameters and conditions. The binding affinity and interactions of the original ligands within co-crystal complexes provided essential metrics for evaluating the docking results of candidate MFH molecules. Refer to Section 2.3.3 for detailed elaboration.
Point 3: Compounds have not been tested as possible PAINS.
Response 3:
Thank you for reviewing our manuscript. We value your insightful suggestion and have taken proactive measures to systematically test our candidates for potential Pan-Assay Interference Compounds (PAINS) characteristics. The contents below have been added in the revised manuscript (highlighted in red), details are as follows:
“3.10 Pan Assay Interference Compounds (PAINS) screening
The pan assay interference compounds (PAINS) screening has evolved into a pivotal element within drug design, the PAINS rule is introduced to identify false positive compounds (frequent hitters) during biological screening initiatives. We obtained a list of known aggregators (12645 molecules were shown in Table S9) from Aggregator Advisor [78], represented the known aggregator molecules and our screened MFH candidates with MACCS fingerprints. We employed RDKit to match the structure of each screened MFH candidate molecule with 12645 known aggregator molecules. As a result, none of MFH candidates (10 potential COX-2 inhibitors and 15 potential mPGES-1 inhibitors) appeared in the known aggregators list.”

Round 2
Reviewer 1 Report
The authors addressed all points raised in the first round of review. Although the lacking experimental validation of the proposed outcomes is still limiting the scientific value, it presents a nice and broadly applicable workflow. The additional docking studies are supporting the findings and partly cope with the lacking experiments.
Author Response
Point: The authors addressed all points raised in the first round of review. Although the lacking experimental validation of the proposed outcomes is still limiting the scientific value, it presents a nice and broadly applicable workflow. The additional docking studies are supporting the findings and partly cope with the lacking experiments.
Response:
We greatly appreciate your thorough review of our manuscript and your constructive feedback. We are pleased to hear that the revisions we made in response to the first round of review were satisfactory. We are encouraged to hear that you find our workflow to be broadly applicable, and we are pleased that the additional docking studies have supported our findings. We understand your concern about the experimental validation, and we acknowledge its importance in strengthening the scientific value of our work. In the future, we will conduct experiments to address this limitation.